# The Impact of Hop Freshness on Kettle-Hopped Beers

**DOI:** 10.3390/foods12234353

**Published:** 2023-12-02

**Authors:** Ksenija Rutnik, Miha Ocvirk, Iztok Jože Košir

**Affiliations:** 1Department for Agrochemistry and Brewing, Slovenian Institute of Hop Research and Brewing, 3310 Žalec, Slovenia; ksenija.rutnik@ihps.si (K.R.); miha.ocvirk@ihps.si (M.O.); 2Biotechnical Faculty, University of Ljubljana, 1000 Ljubljana, Slovenia

**Keywords:** aged hops, beer quality, hop storage index

## Abstract

Hops are an indispensable ingredient in beer, and the differences in their chemical composition impart the various tastes and aromas associated with different beers. However, during storage, hops undergo changes in their chemical composition. Here, the changes in aroma and bitterness of kettle-hopped beers were evaluated in an experiment conducted on three different hop varieties (Aurora, Celeia and Styrian Wolf) with five different hop storage index (HSI) values (0.3–0.7). Hops were added to boiling wort for 5, 45 and 90 min. Alpha-acids, iso-alpha-acids, humulinones, bitterness units and hop aroma compounds in the samples were chemically analysed. All samples also underwent sensorial analysis. The old hops were not problematic in terms of bitterness or early hopping time. However, later additions of old hops reduced the quality and intensity of the hop aroma. The limit value for use without negative consequences for kettle hopping was set at HSI 0.5 for Aurora and Celeia and HSI 0.6 for Styrian Wolf.

## 1. Introduction

Hops (*Humulus lupulus* L.) are a minor ingredient in beer brewing, but they are essential for providing the characteristic flavour and aroma to beers. After harvesting, the chemical composition of hops starts to change; therefore, to obtain the best out of them, hops should ideally be used fresh. However, hops can be harvested only once a year, whereas beer production continues throughout the whole year, making the use of fresh hops impossible for all beers. To our knowledge, only a few research studies have evaluated the impact of the hop storage index (HSI)—an indicator of hop freshness—on beer quality.

Mikyška et al. [1] compared beers brewed with hops of two different varieties with HSI values of 0.31 and 0.46 and 0.29 and 0.43. The beers hopped with aged hops showed an increased value of sulphur dioxide and a slightly increased ratio between higher alcohols and esters. As a result, beer hopped with old hops had a shorter shelf life. The general sensorial impression for beer hopped with fresh hops was slightly higher, but the sensory impact on bitterness and aroma was not evaluated in that study. Srečec et al. [2] conducted a comparison of hops with HSI values of 0.35 and 0.59. Beer hopped with aged hops scored lower on the sensory evaluation, but despite this, the difference was not evaluated as significant. Aged hops also affect the colour of the beer and increase its cost, as the brewing must be prolonged to achieve the same content of iso-alpha-acids in the beer. The most comprehensive investigation thus far, conducted by Rutnik et al. at the Slovenian Institute of Hop Research and Brewing [3], compared three different hop varieties with five different HSI values in terms of chemical and sensorial properties, but they used only a dry hopping technique. We concluded that, with dry hopping, hops with high HSI values lowered the beer quality in terms of aroma and bitterness. We also reported that old hops could induce gushing (over-foaming). We, therefore, set the limit for dry hopping at 0.4 for Aurora and Celeia and at 0.5 for Styrian Wolf. We also prepared another experiment to compare the effects of hop freshness on kettle-hopped beers.

Kettle hopping is a traditional technique and remains the most frequently used hopping technique in the world. In general, three types of kettle hopping are employed: bittering, aroma and continuous hopping [4,5]. The main purpose of bittering hopping is to achieve bitterness in beer. For this, hops have to be added at the beginning of the boil to allow sufficient time for isomerisation of the alpha-acids into iso-alpha-acids, which are the main source of bitterness in beers [6]. Another important reaction for alpha-acids is oxidation, which results in the generation of humulinones. Since these are oxidation products, their levels increase during hop storage. Humulinones are more soluble than iso-alpha-acids due to the presence of an additional hydroxyl group, and they are also 34% less bitter [7]. According to Shellhammer et al. [8] and Rutnik et al. [3], they do not impart a bad quality of bitterness to beer.

A second main purpose is to achieve aroma in beer. For this, the hops must be added late so that the volatiles do not have time to evaporate. The essential oils in the hops are responsible for imparting the hop aroma, and depending on their chemical composition, they can produce a variety of aromas, such as citrus, floral, herbal, hoppy, etc. [9]. During hop storage, the essential oils undergo evaporation and oxidation; consequently, if storage is sufficiently prolonged, the changed chemical composition can result in a completely unexpected aroma imparted into the beer. Essential oils in hops, such as linalool, geraniol and beta-citronellol, contribute to floral and citrus notes. Improper hop storage leads to the autoxidation of myrcene, resulting in undesired aromas. Alpha-humulene, though not often present above the sensory threshold, produces aroma-affecting oxidation products in beer. Humulene epoxide I, resistant to hydrolysis, is the predominant form in beer, contributing to a sophisticated aroma at low concentrations and spicy or mouldy notes at higher concentrations. Beta-caryophyllene, with low solubility, imparts a spicy character, while its oxidation product, caryophyllene oxide, can introduce musty, floral or spicy aromas [10,11,12,13,14,15,16,17,18,19,20,21].

The last hopping regime is the continuous one, which involves adding hops throughout the boiling process and allows for the development of both bittering and aroma properties.

This study aimed to furnish brewers with practical insights into the potential adverse effects of employing aged hops on the quality of both hop aroma and bitterness. Additionally, we identified the HSI threshold beyond which no detrimental effects on the aroma and bitterness of beer were observed.

## 2. Materials and Methods

### 2.1. Chemicals and Standards

Methanol (HPLC grade) was purchased from J. T. Baker (Deventer, The Netherlands); toluene, phosphoric acid (85%) and hydrochloric acid (HCl) (37%) were obtained from Honeywell (Charlotte, NC, USA); 2,2,2-trimethyl pentane (≥99%, ACS reagent), 1-butanol, sodium chloride, sodium hydroxide, ammonium iron (III) citrate, carboxymethyl cellulose (CMC), alpha-pinene (98%), beta-pinene (99%), myrcene (99%), limonene (99%), linalool (97%), alpha-terpineol (99%), beta-citronellol (99%), nerol (98%), geraniol (98%), geranyl acetate (98%), beta-caryophyllene (99%), alpha-humulene (CRM) and caryophyllene oxide (95%) were purchased from Sigma Aldrich (St. Louis, MO, USA); ethylenediaminetetraacetic acid (EDTA) was obtained from Merck (Kenilworth, NJ, USA); international calibration extract 4 (ICE4)—for quantification of alpha-acids and beta-acids—DCHA-Iso (trans-iso-alpha-acids in dycyclohexylamine salt form) and DCHA-Humulinones (trans-humulinones in dycyclohexylamine salt form) were obtained from Labor Veritas (Zürich, Switzerland). Light Malt Extract was procured from Muntons (Lombard, Chicago, IL, USA).

### 2.2. Hop Material

For kettle hopping Celeia, Aurora and Styrian Wolf varieties were chosen. The hop material was supplied by Hmezad Exim d.o.o. from Žalec. The initial HSI was measured immediately, and after that, samples were left to age at an ambient temperature and air. To pick out samples with desirable HSI, every week, the analysis of HSI was made. Samples that reached HSI 0.3, 0.4, 0.5, 0.6 and 0.7 were stored in refrigerator (−18 °C) until experiment. There were five samples of each variety stored, which resulted in 15 samples in total.

### 2.3. Beer Brewing

Wort was prepared from 70 L of water, 4.2 kg of dextrose and 7.5 kg of British Light Extract (Muntons Malted Ingredients, Chicago, IL, USA). The solution was stirred until clear. For brewing, 1.5 L of wort was poured into boiling flask and heated to a boil. At the beginning of boiling, 15 g of hops were added and left to boil for 5 min, 45 min and 90 min (the same sample in three individual flasks). Immediately after boiling, the solution was cooled to 20 °C and filtered through gauze, and the first sample was taken. Then, 1.3 g of yeast (SafAleTM S-04) was added and left to ferment (20 °C) for five days. After fermentation, the excess yeast was removed, another sample was taken, and the beer was prepared for the maturing phase, which lasted for 16 days in a dark place. After the maturing phase, a sample was taken, and the beer was poured into bottles and prepared for sensory evaluation. Samples for sensory evaluation were stored at 4 °C for approximately 1 month. Samples for chemical analysis were stored at −18 °C until analysis. Figure 1 presents flow chart for brewing trials of Celeia, Figure 2 for Aurora and Figure 3 for Styrian Wolf. Further measurements were conducted in duplicate.

### 2.4. Sensory Analysis

Ten trained assessors, 6 males and 4 females, aged 26-55, were evaluating the intensity and quality of hop aroma and bitterness. Training of assessors was carried out as part of FlavorActiv^TM^ training. The method chosen for sensory evaluation followed the method in Analytica EBC 13.13—a routine descriptive test [22]. Briefly, 70 mL of beer, tempered to 10 °C, was served in 250 mL glass. The assessors were asked to evaluate the intensity and quality of bitterness and hop aroma on a scale of 0–5, with 0.5 steps allowed. There were three series of 15 samples, one for each variety. At once, 5 beers with same boiling time and same variety were evaluated for four different characteristics (intensity of bitterness, quality of bitterness, intensity of hop aroma and quality of hop aroma). Altogether, 45 beers were evaluated. The beers were labelled blank, and no discussion was permitted among assessors. One-way ANOVA, followed by a Tukey test (α = 0.05), was performed for statistical processing of the data. Data were analysed using OriginPro^®^ 2020b (OriginLab Corporation, Northampton, MA, USA) software package.

### 2.5. Chemical Analysis

#### 2.5.1. Hop Storage Index

The HSI value was determined according to the Analytica-EBC method 7.13 [23]. A total of 2.5 g of ground hops were weighed into flask, and 25 mL of toluene was added. The flask was placed in a shaker for 45 min. After extraction, 2 mL of filtered solution was diluted with 48 mL of methanol, obtaining solution A. Solution A was further diluted to 25 mL with alkaline methanol, obtaining solution B. The absorbance of solution B was measured spectrophotometrically at 275 nm and 325 nm on Shimadzu UV-1900 UV-VIS spectrophotometer (Shimadzu, Japan) against the blank sample. The ratio between absorbances at those wavelengths provided the HSI value.

#### 2.5.2. Bitterness Units (BUs) of Beer

The BUs of the beer were measured following the official method prescribed by Analytica-EBC 9.8 [24]. To ten millilitres of degassed beer, 0.5 mL of HCl and 20 mL of iso-octane were added. The mixture was shaken for 15 min and centrifuged for 3 min (3000× *g*) after that. The absorbance of the iso-octane layer was measured spectrophotometrically at 275 nm against pure iso-octane on Shimadzu UV-1900 UV-VIS spectrophotometer (Shimadzu, Japan). The BU was calculated according to Equation (1):BU = 50 · A_275_(1)
where A_275_ is the absorbance at 275 nm measured against a reference of pure iso-octane.

The results were rounded to the nearest whole number.

#### 2.5.3. HPLC Analysis of Alpha-Acids, Iso-Alpha-Acids and Humulinones

Hop samples were analysed according to the method in Analytica EBC 7.7 [25]. To 5 g of ground hop, 10 mL of methanol, 20 mL of 0.1 N HCl and 50 mL of diethyl ether were added. The mixture was shaken for 45 min. The ether phase was taken (5 mL) and dissolved in methanol to 50 mL. The solution was filtered through PET filter (Macherey-Nagel, Düren, Germany) into vial and prepared for HPLC-analysis. Beer samples were degassed prior to analysis. A total of 5 mL of sample was poured into glass vial and centrifuged at 3000 rpm for 15 min and filtered through PTFE filters into vials (Macherey-Nagel, Düren, Germany). The columns used for analysis of alpha-acids, humulinones and iso-alpha-acids were Nucleodur^®^ 5-100 C18, 125 × 4 mm (Macherey-Nagel, Düren, Germany). Mobile phase was a solution of 775 mL of distilled water, 210 Ml of methanol and 9 mL of orthophosporic acid. The flow was set to 1 mL/min, the column temperature was 40 °C, and the injection volume was 2 µL. The DAD was set to 270 nm for humulinones and iso-alpha-acids and to 314 nm for alpha-acids. Standards used for quantification were ICE4, DCHA-Iso and DCHA-Humulinones. HPLC analysis was performed with an Agilent 1200 Series (Agilent, Santa Clara, CA, USA) chromatography system, and the data handling was carried out using Agilent ChemStation for LC 3D systems (Rev. B.03.02) [16].

#### 2.5.4. Hop Essential Oil Content

The content of hop essential oil was determined using a steam distillation technique, following the procedure prescribed in Analytica-EBC 7.10 [26]. To 50 g of ground hop, 1 L of distilled water was added. The flask containing the mixture was placed into heating block and attached to the distillation apparatus. After three hours of boiling, the volume of oil in oil trap was measured. The volume of oil was reported on for 100 g of dried sample.

#### 2.5.5. HS-SPME-GC-MS Analysis of Aroma Components Derived from Hops

In 20 mL vials, 10 mL of beer sample, 1 g of sodium chloride (Sigma-Aldrich, Darmstadt, Germany) and 0.5 mL of internal standard iso butanol (Sigma-Aldrich, Darmstadt, Germany) were mixed. The samples were analysed by gas chromatography (Agilent 8890 GC System; Agilent, Santa Clara, CA, USA) coupled with mass spectrometry (597BB GC/MS; Agilent, Santa Clara, CA, USA) using a system equipped with an autosampler (PAL RSI 120; PAL, Zwingen, Switzerland). Agilent MSD ChemStation Enhanced Data Analysis software (Rev. F.01.036.2357) was used for data analysis. The parameters and operating conditions, which followed the method described by Dennenlöhr et al. [27], are listed in Table 1.

For quantification of each compound (myrcene, linalool, alpha-humulene, alpha-terpineol, beta-citronellol, nerol, beta-caryophyllene and caryophyllene-oxide), the calibration curve was constructed. The obtained correlation coefficients were more than 0.99, and the relative standard deviations (RSDs) were as high as ±16%.

## 3. Results

Hops are definitely the ingredient that provides the characteristic taste to every beer made. Since hops contribute to aroma and bitterness, hop quality is crucially important for delivering a desirable final taste. Our goals were to evaluate the impact of aged hops on the bitterness and aroma properties of kettle-brewed beer and to set the limit for the HSI value at which hops are no longer appropriate for brewing.

### 3.1. Hop Analysis

Prior to the experiment, the hops were analysed for their HSI, the content of alpha-acids and humulinones and the content of hop essential oil. The results are summarised in Table 2. With increasing HSI, the content of essential oil and the amount of alpha-acids declined, while the levels of the alpha-acid oxidation product humulinones increased. The standard deviations for the results are presented together with the results in Table 2, except for HSI, where the standard deviation was 0.01 in all cases.

### 3.2. Beer Aroma

For the chemical analysis of beer aroma, we took samples at three stages of beer making: after boiling, after fermentation and after maturation. The sensorial analysis was conducted only on the final sample. The results are presented in the following tables: Table 3 for Celeia samples, Table 4 for Aurora samples and Table 5 for Styrian Wolf samples. Considering the whole process (i.e., samples taken after boiling, fermentation and maturation), myrcene, alpha-humulene and beta-caryophyllene clearly showed a downward trend. This was due to the evaporation of volatiles and their adsorption onto the yeast cells [16]. The monoterpene alcohols underwent numerous biotransformations; in addition to the oxidation and hydration reactions, the yeast cells were also responsible for the formation of new compounds. Ale yeast strains (*S. cerevisiae*) form beta-citronelol, nerol, linalool and alpha-terpineol from geraniol, while linalool is further transformed into nerol and alpha-terpineol [28,29]. Therefore, the linalool and geraniol contents decreased after the fermentation and increased after maturation. By contrast, the content of alpha-terpineol increased after the fermentation and decreased after maturation.

The content of beta-citronelol and nerol increased throughout the entire process, leading us to conclude that only a few or no products are formed from them. The content of humulene epoxide I and humulenol II dropped after fermentation and rose again during maturation. The content of caryophyllene oxide decreased in most of the Aurora and Styrian Wolf samples, whereas the Celeia samples showed an initial increase in caryophyllene oxide, followed by a decrease in some samples. Besides the processing itself, the variety and the age of the hops had an impact on the content of hop essential oil compounds in beer.

Contrary to our expectations, the content of volatiles in the final samples with different boiling times was very comparable, although a difference was noticeable after boiling. The greatest difference was observed for linalool, which decreased in level with increasing boiling time. The lower content of linalool is the consequence of evaporation as well as biotransformation between monoterpene alcohols. A similar difference was noted for myrcene and geraniol as well. Since linalool is one of the most important contributing compounds to beer aroma, we can confirm that the regimen used in kettle hopping is very important.

The differences in the contents of volatiles were smaller than those observed after dry hopping [3] and were most noticeable during sensorial evaluation. The samples that were boiled for 5 min contained a higher amount of myrcene only for the samples with HSI 0.3; otherwise, the myrcene content was the same regardless of the age of the hop. In Celeia, the content of myrcene in samples boiled longer than 5 min was below the LOD (19.35 µg/L). In Aurora and Styrian Wolf, the levels were higher, and the amounts decreased with increasing HSI until a certain HSI was reached, for which the difference was no longer significant. The linalool content increased and decreased depending on the variety and boiling duration. These findings proved that, during kettle hopping, volatiles are transferred into the beer, but many conversions and biotransformations also occur [28,29,30]. The difference observed for the alpha-terpineol content with increasing HSI was not statistically significant, but the levels were above the sensorial threshold, meaning that alpha-terpineol did contribute to aroma. As observed with the dry hopping technique, the contents of beta-citronelol and nerol did not follow any clear pattern [3]. The geraniol content declined with increasing HSI and then plateaued at a certain point. Despite the drop in content, the level of geraniol was still above the sensorial threshold. The same pattern was observed for alpha-humulene and beta-caryophyllene, but the levels were below the sensorial threshold, meaning that these compounds made no contribution to the beer aroma. The content of caryophyllene oxide decreased and stabilised for Styrian Wolf and Aurora but increased in the Celeia variety. Caryophyllene oxide is an oxidation product; therefore, we expected to see the same behaviour for Styrian Wolf and Aurora. The contradiction could reflect a further hydrolysis of caryophyllene oxide to seven new products, with clovane-diol as a major one. Clovane-diol can impart a spicy character to beer [31]. The content of the other two oxidation products, humulene epoxide II and humulinone II, increased with the ageing of the hops.

The chemical analysis results showed differences between individual brewing times and hop ages that were smaller than expected, but the beer aroma is the result of the contents and mutual effects of many components that cannot be measured by chemical analysis. For this reason, a sensorial evaluation was needed. We evaluated the quality and intensity of the hop aroma in all samples. The results are shown in Table 6 and are presented as mean value ± standard deviation. The results were analysed statistically by one-way ANOVA, followed by the Tukey test. Figure 4 shows the data for the quality of hop aroma, and Figure 5 shows the intensity of the hop aroma. The x-axis shows the mean difference between the samples, and the y-axis shows the HSIs of the compared samples. The level of confidence α is 0.05.

Ageing of the hops (increasing HSI) caused a decrease in the quality of the hop aroma in all samples. In beers brewed with hops with an HSI of 0.3, the quality of hop aroma was statistically higher than in beers brewed with hops with an HSI of 0.5. In Aurora and Styrian Wolf, the quality of hop aroma decreased with increasing boiling time, meaning that compounds contributing to typical beer aroma were no longer present in the beer. The difference between samples with the same HSI but different boiling times was also greater for old hops, meaning that undesirable compounds contribute to the aroma of beers brewed with old hops. Interestingly, the same pattern was not observed in the Celeia variety, as the Celeia samples produced the highest aroma quality if boiled for 45 min, regardless of the HSI. The reason for this must be the presence of some as-yet-unmeasured compounds or the occurrence of some as-yet-unidentified synergistic effect between the numerous compounds present in the hops. Another interesting observation, as seen in the Tukey’s plots, was that Aurora and Celeia samples with the same boiling time differed less when longer boiling times were applied. This means that extending the boiling time will cause the loss of some desirable compounds but will also hide some irregularities in the hoppy beer aroma. The Styrian Wolf results did not hold to this pattern, as all samples brewed with hops with HSI higher than 0.4 were statistically different. The only exceptions were samples with HSI 0.6 and 0.7 boiled for 5 min. Besides humulene epoxide I and humulenol II, which are oxidation products and whose content increases with increasing HSI, we suppose that some other products of ageing must be forming since we noticed the smell of aged and oxygenated hops in beers brewed with high HSI. The intensity of aroma also decreased with increasing HSI. Despite the fact that the content of the investigated compounds did not decrease in all cases, the intensity of hop aroma steadily decreased. This is another confirmation that beer aroma is formed based on the synergistic effects of numerous compounds and cannot be described based on only a few otherwise very important compounds. The highest intensity of aroma was found in samples boiled for only 5 min with HSI 0.3. Samples brewed with hops with HSI higher than 0.5 had very low aroma intensities and almost unnoticeable aromas in samples boiled for a long time. For the best aroma, we would recommend setting the limit for kettle hopping for Celeia and Aurora at around 0.5, especially because of the intensity drop. The limit for Styrian Wolf is an HSI of 0.6. Styrian Wolf did better in the dry hopping experiment; therefore, it is no surprise that the limit for Styrian Wolf was also higher in kettle hopping [3].

### 3.3. Beer Bitterness

Table 7 presents the results of the chemical analysis of beer bitterness. The results are presented as mean value ± standard deviation.

Comparing the samples brewed with the same variety but with different boiling times, the levels of iso-alpha-acids were higher in those that were boiled the longest. Consequently, in most cases, the measured bitterness was also higher. The levels of humulinones decreased slightly in samples with the same HSI and different boiling times. From that, we can conclude that humulinones are less thermally stable than iso-alpha-acids, which need high temperatures to form. The alpha-acid content in beers boiled for 90 min was almost zero, meaning that all the alpha-acids transferred into the wort had isomerised or transformed into other compounds. We also noticed that the ratio of isomerisation depended on the hop variety since a higher percentage was transformed in the Celeia variety than in the Aurora and Styrian Wolf varieties.

Increasing the HSI caused a noticeable decrease in alpha-acids that correlated with the alpha-acid content in the initial sample, where their levels also decreased with increasing HSI. Consistent with the lower alpha-acid content, the iso-alpha-acid content was also lower, but the degree of isomerisation was not correlated with the HSI. All the measured compounds absorb at a wavelength of 275 nm; therefore, we expected that the pattern of changes in measured bitterness would somehow follow the sum of the bitterness compounds. This expectation was not met, but in most cases, we noticed that the highest values for measured bitterness were at an HSI of 0.5. At the same time, we noticed that the humulinone levels rose with increases in HSI to a certain level but then stabilised or started to drop. This usually happened at an HSI of 0.5, similar to the pattern observed with dry hopping [3]. Since the humulinones content in hop samples increased with rising HSI, some transformation must occur from humulinones to some other products during the beer brewing process. We propose that the formed compounds are 4′-hydroxyallo-humulinones, which were discovered by Taniguchi et al. in 2013 [32]. The 4′-hydroxyallo-humulinones are oxidation products of humulinones and form part of the hard fraction of hop resins, which are insoluble in hexane. The sensorial impact of 4′-hydroxyallo-humulinones has not been evaluated yet.

In addition to the chemical analysis of beer bitterness, a sensorial evaluation was also performed, but only for Celeia and Styrian Wolf because Aurora was too bitter to obtain proper results. The results are presented as mean value ± standard deviations in Table 8. Figure 6 and Figure 7 present the Tukey’s plots for the quality and intensity of bitterness for the Celeia and Styrian Wolf results.

The quality of bitterness after 5 min of boiling decreased with increasing HSI, and the samples were statistically different between HSI 0.4 and HSI 0.6. With longer boiling times, the quality of bitterness in Celeia beers increased, and beers brewed with hops with HSI 0.7 have a statistically higher quality of bitterness than beers brewed with hops with HSI 0.3. The same pattern was only observed for Styrian Wolf beers boiled for 90 min. Even then, an exception was noted, as the beer with HSI 0.5 scored the highest. That beer had the highest levels of humulinones and the lowest levels of alpha-acids; therefore, we can conclude that humulinones do not result in bad-quality bitterness. The intensity of bitterness in samples boiled for 5 min increased, but only samples with HSI 0.4 and 0.5 were statistically different. At longer boiling times (45 min and 90 min), the intensity of bitterness decreased with higher HSI, and once again, only samples between two HSI levels were statistically different. From this, we can conclude that aged hops have a greater impact on the quality of bitterness than on the intensity of bitterness. Comparing the chemical analysis of bitterness with the sensorial analysis revealed no correlation, meaning that sensorial evaluation is very important. From these results, we can also conclude that the presence of non-isomerised alpha-acids can impact the quality of bitterness since samples with no alpha-acids received higher grades. To summarise, if boiling is conducted for a sufficient time, old hops are not problematic in terms of delivering a bad quality or intensity of bitterness, but they are certainly not suitable for late dosage uses.

## 4. Conclusions

This study evaluated the impact of aged hops on the quality and intensity of beer aroma and bitterness in kettle-hopped beers. If boiling is conducted for a long enough time (at least 45 min), then aged hops are not problematic from the point of bitterness since the negative aftertaste caused by oxidation products is masked by iso-alpha-acids. However, the quality and the intensity of the hop beer aroma are affected, and these decrease as HSI increases. Increasing the boiling time can decrease the difference between samples hopped with hops of different HSI values, suggesting that extending the boiling time causes a loss of some desirable compounds while also hiding some irregularities in the hoppy beer aroma. In addition to humulene epoxide I and humulenol II, which increased in content with the age of hops in the beer samples, we conclude that other ageing products were formed, given the sensory evaluation of beers with a high HSI of aged and oxygenated hops. Based on the results, we have estimated that Celeia and Aurora hops, with an HSI of up to 0.5 and Styrian Wolf, with an HSI of up to 0.6, are suitable for beer brewing.

## Figures and Tables

**Figure 1 foods-12-04353-f001:**
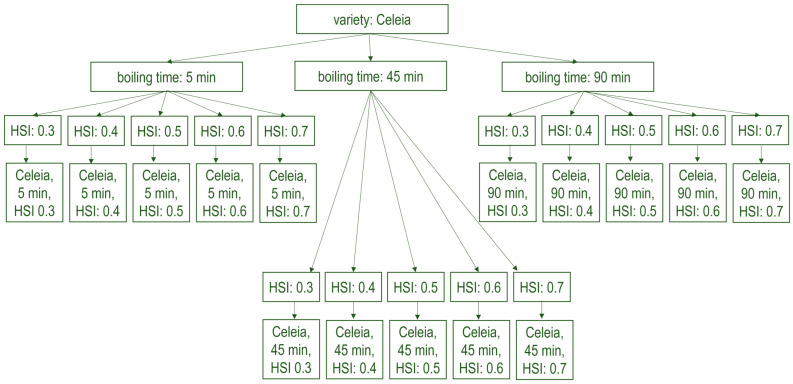
Brewing trials for Celeia variety.

**Figure 2 foods-12-04353-f002:**
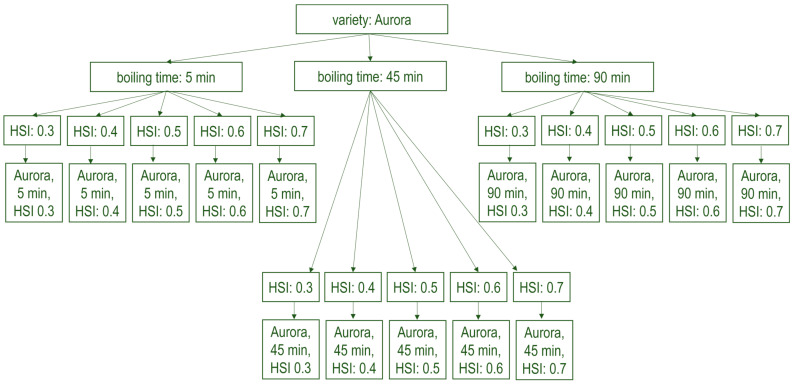
Brewing trials for Aurora variety.

**Figure 3 foods-12-04353-f003:**
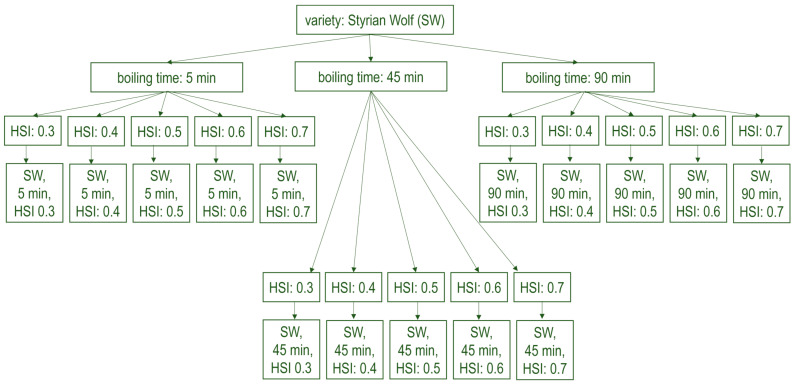
Brewing trials for Styrian Wolf variety.

**Figure 4 foods-12-04353-f004:**
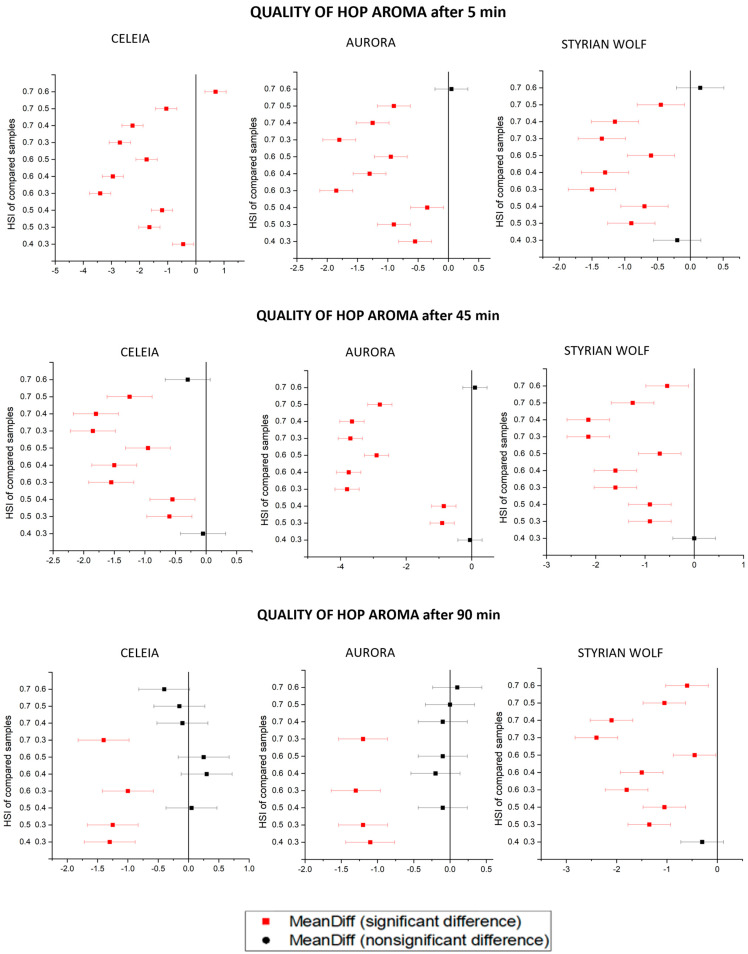
Tukey’s plots for quality of hop aroma for all samples.

**Figure 5 foods-12-04353-f005:**
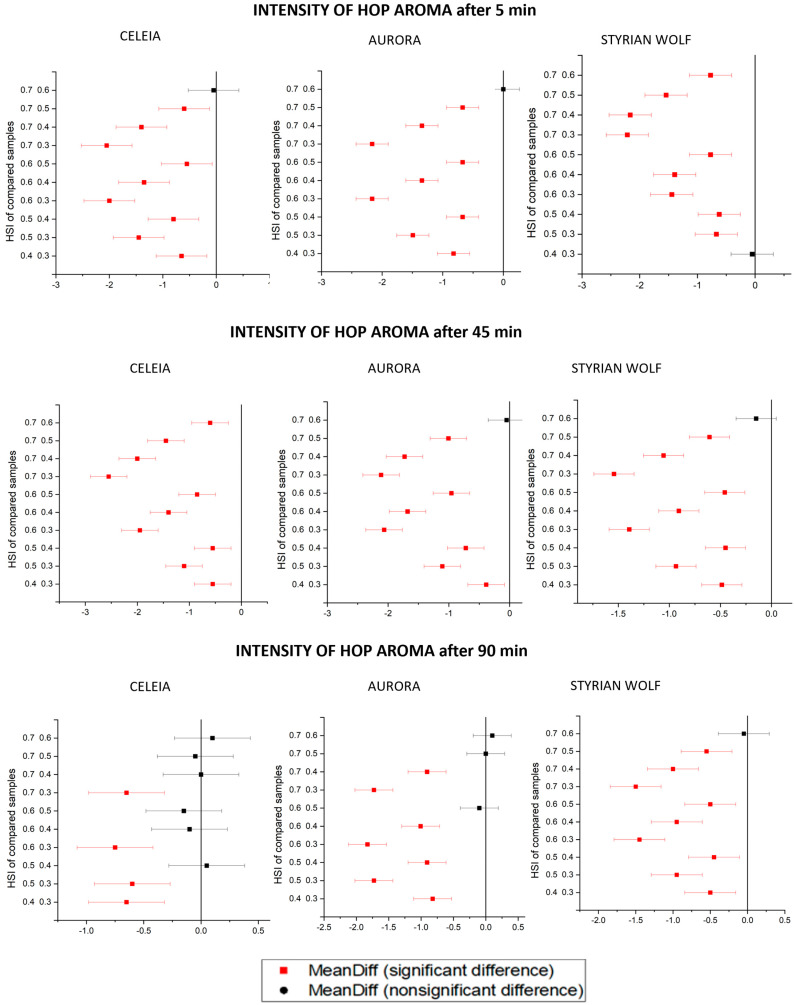
Tukey’s plots for intensity of hop aroma for all samples.

**Figure 6 foods-12-04353-f006:**
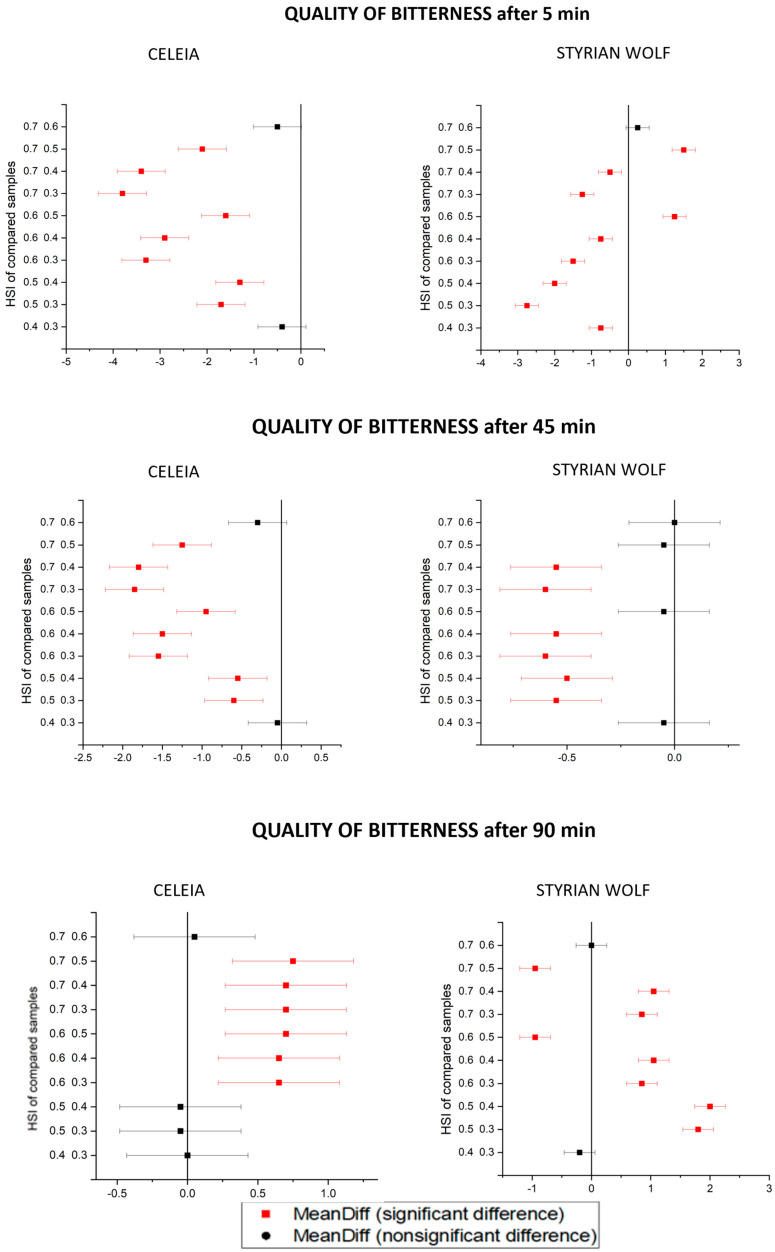
Tukey’s plots for quality of bitterness.

**Figure 7 foods-12-04353-f007:**
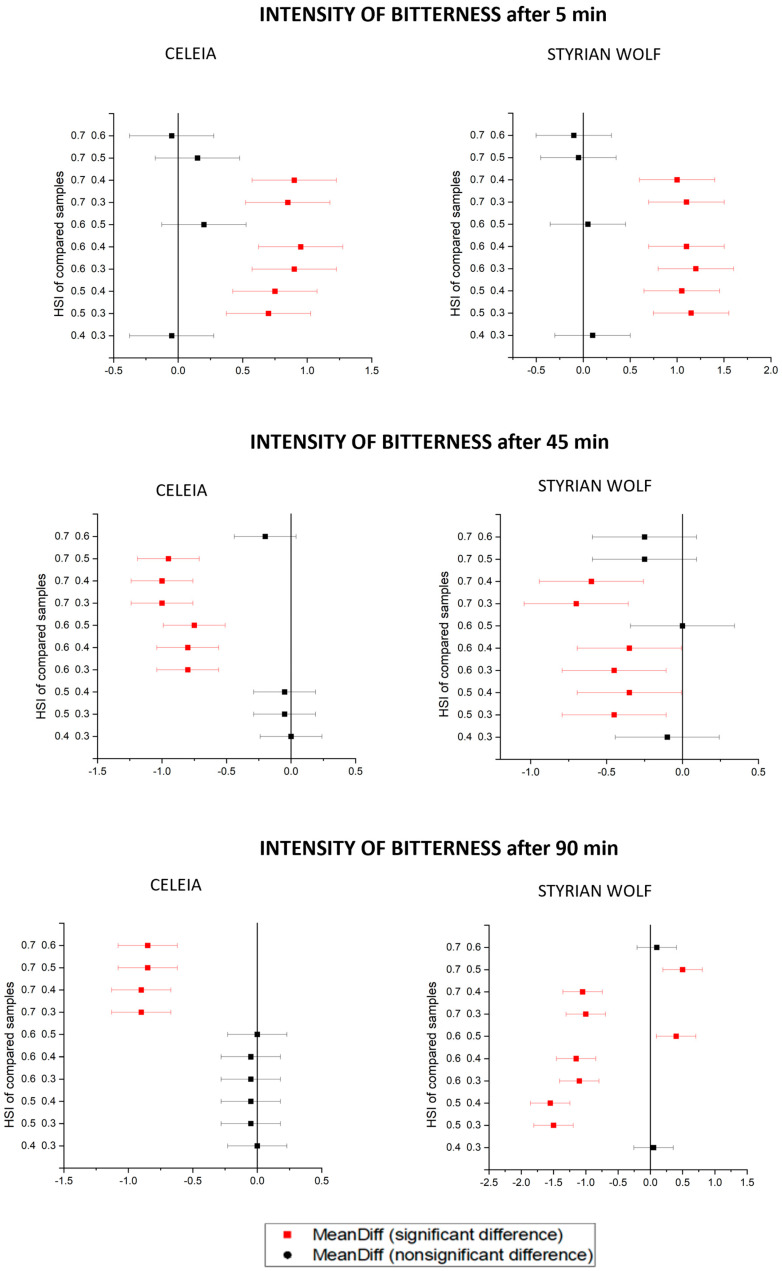
Tukey’s plot for intensity of bitterness.

**Table 1 foods-12-04353-t001:** Operating conditions for HS-SPME-GS-MS determinations of hop volatiles in beer.

Parameter	Value
SPME fibre	50/30 μm DVB/CAR/PDMS (Supelco, Bellefonte, PA, USA)
Incubation time	7.5 min
Extraction time	20 min
Incubation temperature	60 °C
Agitation rate	500 rpm
Pre-desorption time	20 min
Desorption temperature	250 °C
Liner	Ultra Inert SPME Liner 0.75 mm
Split ratio	1:20
GC column	HP-5MS UI (30 m × 250 μm × 0.25 μm) (Agilent, Santa Clara, CA, USA)
Mobile phase	Helium (purity 5.0)
Temperature program	50 °C–190 °C (10 °C/min) 190 °C–300 °C (70 °C/min), 1 min hold
Ion source temperature	230 °C
MS quad temperature	150 °C
Gain	1.000
Acquisition mode	SIM and SCAN

**Table 2 foods-12-04353-t002:** Results of hop analysis.

Variety	HSI	Essential Oil [mL/100 g]	Alpha-Acids [%]	Humulinones [%]
Celeia	0.3	1.05 ± 0.04	3.19 ± 0.05	0.11 ± 0.01
Celeia	0.4	0.89 ± 0.03	2.38 ± 0.03	0.15 ± 0.01
Celeia	0.5	0.85 ± 0.03	2.06 ± 0.03	0.17 ± 0.01
Celeia	0.6	0.74 ± 0.03	1.80 ± 0.03	0.18 ± 0.01
Celeia	0.7	0.46 ± 0.02	1.60 ± 0.02	0.18 ± 0.01
Aurora	0.3	1.97 ± 0.08	12.36 ± 0.18	0.43 ± 0.03
Aurora	0.4	0.98 ± 0.04	10.38 ± 0.15	0.49 ± 0.03
Aurora	0.5	0.79 ± 0.03	8.36 ± 0.12	0.64 ± 0.04
Aurora	0.6	0.51 ± 0.02	7.36 ± 0.11	0.65 ± 0.04
Aurora	0.7	0.44 ± 0.02	6.05 ± 0.09	0.69 ± 0.04
Styrian Wolf	0.3	2.22 ± 0.09	11.87 ± 0.17	0.42 ± 0.03
Styrian Wolf	0.4	1.49 ± 0.06	8.66 ± 0.13	0.54 ± 0.03
Styrian Wolf	0.5	1.28 ± 0.05	9.21 ± 0.13	0.60 ± 0.04
Styrian Wolf	0.6	1.07 ± 0.04	7.71 ± 0.11	0.60 ± 0.04
Styrian Wolf	0.7	1.02 ± 0.04	7.66 ± 0.11	0.72 ± 0.05

**Table 3 foods-12-04353-t003:** Content of hop aroma compounds in beer samples brewed with Celeia variety determined by HS-SPME-GC-MS.

Variety	HSI	Boiling Time	Sampling	Myrcene [μg/L]	Linalool [μg/L]	Alpha-Terpineol [μg/L]	Beta-Citronellol [μg/L]	Geraniol [μg/L]	Beta-Caryophyllene [μg/L]	Alpha-Humulene [μg/L]	Caryophyllene Oxide [μg/L]	Humulene Epoxide I [rel. %]	Humulenol II [rel. %]
Celeia	0.3	5	boiling	2956 ± 473 ^0^	366 ± 32 ^0^	44 ± 8 ^0^	31 ± 2 ^0^	131 ± 6 ^0^	850 ± 34 ^0^	385 ± 60 ^0^	50 ± 5 ^0^	0.21 ± 0.01 ^0^	0.96 ± 0.06 ^0^
Celeia	0.4	5	boiling	2427 ± 388 ^1^	345 ± 30 ^0^	41 ± 7 ^0^	12 ± 1 ^1^	107 ± 5 ^1^	659 ± 27 ^1^	293 ± 46 ^1^	118 ± 13 ^1^	0.61 ± 0.04 ^1^	2.51 ± 0.16 ^1^
Celeia	0.5	5	boiling	1805 ± 289 ^1^	443 ± 39 ^1^	62 ± 11 ^1^	15 ± 1 ^1^	103 ± 4 ^1^	599 ± 24 ^1^	304 ± 48 ^1^	197 ± 21 ^1^	0.97 ± 0.06 ^1^	3.79 ± 0.25 ^1^
Celeia	0.6	5	boiling	1009 ± 161 ^1^	403 ± 36 ^1^	55 ± 10 ^1^	15 ± 1 ^1^	122 ± 5 ^1^	479 ± 19 ^1^	263 ± 41 ^1^	219 ± 23 ^1^	1.33 ± 0.09 ^1^	5.21 ± 0.34 ^1^
Celeia	0.7	5	boiling	1047 ± 167 ^1^	429 ± 38 ^1^	60 ± 11 ^1^	15 ± 1 ^1^	93 ± 4 ^1^	477 ± 19 ^1^	250 ± 39 ^1^	187 ± 20 ^1^	1.3 ± 0.08 ^1^	5.39 ± 0.35 ^1^
Celeia	0.3	45	boiling	314 ± 50 ^0^	227 ± 20 ^0^	22 ± 4 ^0^	<LOD ^0^	107 ± 5 ^0^	691 ± 28 ^0^	338 ± 53 ^0^	69 ± 7 ^0^	0.33 ± 0.02 ^0^	1.39 ± 0.09 ^0^
Celeia	0.4	45	boiling	259 ± 41 ^0^	280 ± 25 ^1^	32 ± 6 ^1^	<LOD ^0^	122 ± 5 ^1^	748 ± 30 ^1^	347 ± 54 ^0^	163 ± 17 ^1^	0.77 ± 0.05 ^1^	2.54 ± 0.16 ^1^
Celeia	0.5	45	boiling	193 ± 31 ^0^	249 ± 22 ^1^	25 ± 4 ^1^	<LOD ^0^	82 ± 4 ^1^	358 ± 14 ^1^	203 ± 32 ^1^	156 ± 17 ^1^	1.16 ± 0.08 ^1^	4.06 ± 0.26 ^1^
Celeia	0.6	45	boiling	200 ± 32 ^0^	270 ± 24 ^1^	30 ± 5 ^1^	12 ± 1 ^1^	90 ± 4 ^1^	450 ± 18 ^1^	250 ± 39 ^1^	212 ± 23 ^1^	1.47 ± 0.10 ^1^	4.63 ± 0.3 ^1^
Celeia	0.7	45	boiling	211 ± 34 ^0^	360 ± 32 ^1^	44 ± 8 ^1^	14 ± 1 ^1^	98 ± 4 ^1^	763 ± 31 ^1^	350 ± 55 ^1^	241 ± 26 ^1^	1.22 ± 0.08 ^1^	4.48 ± 0.29 ^1^
Celeia	0.3	90	boiling	268 ± 43 ^0^	259 ± 23 ^0^	35 ± 6 ^0^	<LOD ^0^	98 ± 4 ^0^	973 ± 39 ^0^	445 ± 70 ^0^	92 ± 10 ^0^	0.43 ± 0.03 ^0^	1.45 ± 0.09 ^0^
Celeia	0.4	90	boiling	146 ± 23 ^1^	228 ± 20 ^1^	26 ± 5 ^1^	15 ± 1 ^1^	93 ± 4 ^1^	276 ± 11 ^1^	150 ± 23 ^1^	106 ± 11 ^1^	1.03 ± 0.07 ^1^	3.7 ± 0.24 ^1^
Celeia	0.5	90	boiling	129 ± 21 ^1^	244 ± 21 ^1^	30 ± 5 ^1^	12 ± 1 ^1^	87 ± 4 ^1^	324 ± 13 ^1^	202 ± 32 ^1^	177 ± 19 ^1^	1.42 ± 0.09 ^1^	4.44 ± 0.29 ^1^
Celeia	0.6	90	boiling	131 ± 21 ^1^	185 ± 16 ^1^	25 ± 4 ^1^	15 ± 1 ^1^	63 ± 3 ^1^	256 ± 10 ^1^	148 ± 23 ^1^	145 ± 15 ^1^	1.52 ± 0.10 ^1^	5.66 ± 0.37 ^1^
Celeia	0.7	90	boiling	147 ± 24 ^1^	294 ± 26 ^1^	39 ± 7 ^1^	14 ± 1 ^1^	102 ± 4 ^1^	558 ± 22 ^1^	286 ± 45 ^1^	221 ± 24 ^1^	1.46 ± 0.09 ^1^	4.98 ± 0.32 ^1^
Celeia	0.3	5	fermentation	163 ± 26 ^0^	281 ± 25 ^0^	391 ± 69 ^0^	29 ± 2 ^0^	55 ± 2 ^0^	375 ± 15 ^0^	183 ± 29 ^0^	10 ± 1 ^0^	0.10 ± 0.01 ^0^	0.18 ± 0.01 ^0^
Celeia	0.4	5	fermentation	123 ± 20 ^1^	251 ± 22 ^0^	315 ± 56 ^0^	27 ± 2 ^0^	42 ± 2 ^1^	210 ± 8 ^1^	115 ± 18 ^1^	34 ± 4 ^1^	0.30 ± 0.02 ^1^	0.6 ± 0.04 ^1^
Celeia	0.5	5	fermentation	93 ± 15 ^1^	288 ± 25 ^0^	369 ± 65 ^0^	42 ± 3 ^1^	35 ± 2 ^1^	80 ± 3 ^1^	49 ± 8 ^1^	48 ± 5 ^1^	0.59 ± 0.04 ^1^	0.83 ± 0.05 ^1^
Celeia	0.6	5	fermentation	81 ± 13 ^1^	325 ± 29 ^1^	347 ± 61 ^0^	33 ± 3 ^1^	49 ± 2 ^1^	170 ± 7 ^1^	107 ± 17 ^1^	75 ± 8 ^1^	0.59 ± 0.04 ^1^	1.41 ± 0.09 ^1^
Celeia	0.7	5	fermentation	101 ± 16 ^1^	354 ± 31 ^1^	364 ± 64 ^0^	35 ± 3 ^1^	40 ± 2 ^1^	177 ± 7 ^1^	115 ± 18 ^1^	65 ± 7 ^1^	0.58 ± 0.04 ^1^	1.6 ± 0.1 ^1^
Celeia	0.3	45	fermentation	59 ± 9 ^0^	205 ± 18 ^0^	419 ± 74 ^0^	31 ± 2 ^0^	43 ± 2 ^0^	289 ± 12 ^0^	179 ± 28 ^0^	31 ± 3 ^0^	0.19 ± 0.01 ^0^	0.4 ± 0.03 ^0^
Celeia	0.4	45	fermentation	61 ± 10 ^0^	240 ± 21 ^1^	414 ± 73 ^0^	31 ± 2 ^0^	47 ± 2 ^1^	324 ± 13 ^1^	195 ± 31 ^1^	71 ± 8 ^1^	0.44 ± 0.03 ^1^	0.9 ± 0.06 ^1^
Celeia	0.5	45	fermentation	59 ± 9 ^0^	220 ± 19 ^1^	449 ± 79 ^1^	41 ± 3 ^1^	42 ± 2 ^1^	167 ± 7 ^1^	117 ± 18 ^1^	71 ± 8 ^1^	0.61 ± 0.04 ^1^	1.17 ± 0.08 ^1^
Celeia	0.6	45	fermentation	67 ± 11 ^1^	261 ± 23 ^1^	375 ± 66 ^1^	51 ± 4 ^1^	39 ± 2 ^1^	237 ± 10 ^1^	162 ± 25 ^1^	70 ± 7 ^1^	0.75 ± 0.05 ^1^	1.55 ± 0.10 ^1^
Celeia	0.7	45	fermentation	66 ± 11 ^1^	275 ± 24 ^1^	393 ± 69 ^1^	49 ± 4 ^1^	39 ± 2 ^1^	293 ± 12 ^1^	196 ± 31 ^1^	94 ± 10 ^1^	0.76 ± 0.05 ^1^	1.98 ± 0.13 ^1^
Celeia	0.3	90	fermentation	70 ± 11 ^0^	219 ± 19 ^0^	461 ± 81 ^0^	27 ± 2 ^0^	44 ± 2 ^0^	301 ± 12 ^0^	186 ± 29 ^0^	25 ± 3 ^0^	0.19 ± 0.01 ^0^	0.4 ± 0.03 ^0^
Celeia	0.4	90	fermentation	52 ± 8 ^1^	188 ± 17 ^1^	481 ± 85 ^0^	24 ± 2 ^1^	41 ± 2 ^1^	93 ± 4 ^1^	62 ± 10 ^1^	33 ± 4 ^1^	0.31 ± 0.02 ^1^	0.85 ± 0.06 ^1^
Celeia	0.5	90	fermentation	50 ± 8 ^1^	192 ± 17 ^1^	454 ± 80 ^0^	40 ± 3 ^1^	36 ± 2 ^1^	116 ± 5 ^1^	87 ± 14 ^1^	54 ± 6 ^1^	0.5 ± 0.03 ^1^	1.31 ± 0.08 ^1^
Celeia	0.6	90	fermentation	49 ± 8 ^1^	155 ± 14 ^1^	402 ± 71 ^1^	29 ± 2 ^1^	36 ± 2 ^1^	89 ± 4 ^1^	63 ± 10 ^1^	45 ± 5 ^1^	0.45 ± 0.03 ^1^	1.57 ± 0.10 ^1^
Celeia	0.7	90	fermentation	88 ± 14 ^1^	265 ± 23 ^1^	451 ± 80 ^1^	47 ± 4 ^1^	45 ± 2 ^1^	200 ± 8 ^1^	143 ± 22 ^1^	73 ± 8 ^1^	0.73 ± 0.05 ^1^	2.07 ± 0.13 ^1^
Celeia	0.3	5	maturation	123 ± 20 ^0^	420 ± 37 ^0^	352 ± 62 ^0^	33 ± 3 ^0^	73 ± 3 ^0^	55 ± 2 ^0^	28 ± 4 ^0^	32 ± 3 ^0^	0.17 ± 0.01 ^0^	0.23 ± 0.01 ^0^
Celeia	0.4	5	maturation	23 ± 4 ^1^	364 ± 32 ^0^	315 ± 56 ^0^	37 ± 3 ^1^	63 ± 3 ^1^	51 ± 2 ^1^	28 ± 4 ^0^	41 ± 4 ^0^	0.58 ± 0.04 ^1^	0.78 ± 0.05 ^1^
Celeia	0.5	5	maturation	21 ± 3 ^1^	500 ± 44 ^1^	329 ± 58 ^0^	47 ± 4 ^1^	65 ± 3 ^1^	50 ± 2 ^1^	30 ± 5 ^0^	80 ± 9 ^1^	0.91 ± 0.06 ^1^	1.15 ± 0.07 ^1^
Celeia	0.6	5	maturation	<LOD ^1^	481 ± 42 ^1^	334 ± 59 ^0^	51 ± 4 ^1^	65 ± 3 ^1^	62 ± 3 ^1^	38 ± 6 ^1^	98 ± 10 ^1^	1.22 ± 0.08 ^1^	1.92 ± 0.12 ^1^
Celeia	0.7	5	maturation	22 ± 4 ^1^	533 ± 47 ^1^	298 ± 53 ^0^	59 ± 5 ^1^	58 ± 2 ^1^	60 ± 2 ^1^	38 ± 6 ^1^	86 ± 9 ^1^	1.31 ± 0.08 ^1^	2.48 ± 0.16 ^1^
Celeia	0.3	45	maturation	<LOD ^0^	297 ± 26 ^0^	330 ± 58 ^0^	30 ± 2 ^0^	63 ± 3 ^0^	49 ± 2 ^0^	31 ± 5 ^0^	15 ± 2 ^0^	0.28 ± 0.02 ^0^	0.44 ± 0.03 ^0^
Celeia	0.4	45	maturation	<LOD ^0^	305 ± 27 ^0^	350 ± 62 ^0^	41 ± 3 ^1^	55 ± 2 ^1^	58 ± 2 ^1^	26 ± 4 ^1^	58 ± 6 ^1^	0.91 ± 0.06 ^1^	1.19 ± 0.08 ^1^
Celeia	0.5	45	maturation	<LOD ^0^	326 ± 29 ^1^	370 ± 65 ^1^	42 ± 3 ^1^	56 ± 2 ^1^	24 ± 1 ^1^	15 ± 2 ^1^	79 ± 8 ^1^	1.06 ± 0.07 ^1^	1.72 ± 0.11 ^1^
Celeia	0.6	45	maturation	<LOD ^0^	367 ± 32 ^1^	231 ± 41 ^1^	62 ± 5 ^1^	56 ± 2 ^1^	30 ± 1 ^1^	19 ± 3 ^1^	76 ± 8 ^1^	1.51 ± 0.10 ^1^	2.98 ± 0.19 ^1^
Celeia	0.7	45	maturation	<LOD ^0^	407 ± 36 ^1^	291 ± 51 ^1^	65 ± 5 ^1^	58 ± 3 ^1^	27 ± 1 ^1^	11 ± 2 ^1^	95 ± 10 ^1^	1.62 ± 0.10 ^1^	3.43 ± 0.22 ^1^
Celeia	0.3	90	maturation	<LOD ^0^	268 ± 24 ^0^	339 ± 60 ^0^	29 ± 2 ^0^	58 ± 2 ^0^	62 ± 2 ^0^	37 ± 6 ^0^	15 ± 2 ^0^	0.34 ± 0.02 ^0^	0.48 ± 0.03 ^0^
Celeia	0.4	90	maturation	26 ± 4 ^1^	243 ± 21 ^1^	309 ± 54 ^1^	32 ± 2 ^1^	45 ± 2 ^1^	21 ± 1 ^1^	11 ± 2 ^1^	32 ± 3 ^1^	0.71 ± 0.05 ^1^	1.4 ± 0.09 ^1^
Celeia	0.5	90	maturation	<LOD ^1^	296 ± 26 ^1^	372 ± 66 ^1^	42 ± 3 ^1^	56 ± 2 ^1^	23 ± 1 ^1^	14 ± 2 ^1^	58 ± 6 ^1^	1.02 ± 0.07 ^1^	2.04 ± 0.13 ^1^
Celeia	0.6	90	maturation	<LOD ^1^	207 ± 18 ^1^	313 ± 55 ^1^	51 ± 4 ^1^	39 ± 2 ^1^	14 ± 1 ^1^	7 ± 1 ^1^	46 ± 5 ^1^	1.72 ± 0.11 ^1^	3.72 ± 0.24 ^1^
Celeia	0.7	90	maturation	<LOD ^1^	367 ± 32 ^1^	347 ± 61 ^1^	55 ± 4 ^1^	56 ± 2 ^1^	15 ± 1 ^1^	4 ± 1 ^1^	77 ± 8 ^1^	1.85 ± 0.12 ^1^	4.12 ± 0.27 ^1^

The analysis of variance (ANOVA) was carried out at 0.05 significance level in order to differentiate between samples brewed with fresh hops with HSI 0.3 (0) and other samples with significant differences (1). LOD: limit of detection.

**Table 4 foods-12-04353-t004:** Content of hop aroma compounds in beer samples brewed with Aurora variety determined by HS-SPME-GC-MS.

Variety	HSI	Boiling Time	Sampling	Myrcene [μg/L]	Linalool [μg/L]	Alpha-Terpineol [μg/L]	Beta-Citronellol [μg/L]	Geraniol [μg/L]	Beta-Caryophyllene [μg/L]	Alpha-Humulene. [μg/L]	Caryophyllene Oxide [μg/L]	Humulene Epoxide I [rel. %]	Humulenol II [rel. %]
Aurora	0.3	5	boiling	2186 ± 350 ^0^	852 ± 75 ^0^	108 ± 19 ^0^	27 ± 2 ^0^	271 ± 12 ^0^	405 ± 16 ^0^	270 ± 42 ^0^	25 ± 3 ^0^	0.27 ± 0.02 ^0^	1.20 ± 0.08 ^0^
Aurora	0.4	5	boiling	1894 ± 303 ^1^	503 ± 44 ^1^	98 ± 17 ^0^	18 ± 1 ^1^	169 ± 7 ^1^	306 ± 12 ^1^	217 ± 34 ^0^	35 ± 4 ^1^	0.47 ± 0.03 ^0^	2.06 ± 0.13 ^0^
Aurora	0.5	5	boiling	701 ± 112 ^1^	449 ± 40 ^1^	86 ± 15 ^0^	18 ± 1 ^1^	141 ± 6 ^1^	130 ± 5 ^1^	106 ± 17 ^1^	38 ± 4 ^1^	1.07 ± 0.07 ^1^	4.87 ± 0.32 ^1^
Aurora	0.6	5	boiling	609 ± 97 ^1^	589 ± 52 ^1^	120 ± 21 ^0^	19 ± 1 ^1^	151 ± 6 ^1^	129 ± 5 ^1^	119 ± 19 ^1^	14 ± 2 ^1^	1.30 ± 0.08 ^1^	5.47 ± 0.35 ^1^
Aurora	0.7	5	boiling	334 ± 53 ^1^	568 ± 50 ^1^	103 ± 18 ^0^	19 ± 1 ^1^	144 ± 6 ^1^	77 ± 3 ^1^	75 ± 12 ^1^	11 ± 1 ^1^	1.62 ± 0.1 ^1^	7.34 ± 0.48 ^1^
Aurora	0.3	45	boiling	3164 ± 506 ^0^	517 ± 46 ^0^	119 ± 21 ^0^	20 ± 1 ^0^	183 ± 8 ^0^	832 ± 34 ^0^	472 ± 74 ^0^	33 ± 4 ^0^	0.23 ± 0.01 ^0^	0.84 ± 0.05 ^0^
Aurora	0.4	45	boiling	860 ± 137 ^1^	383 ± 34 ^1^	78 ± 14	17 ± 1 ^1^	151 ± 6 ^1^	408 ± 16 ^1^	282 ± 44 ^1^	38 ± 4 ^1^	0.66 ± 0.04 ^1^	2.47 ± 0.16 ^1^
Aurora	0.5	45	boiling	294 ± 47 ^1^	271 ± 24 ^1^	49 ± 9	16 ± 1 ^1^	108 ± 5 ^1^	103 ± 4 ^1^	86 ± 13 ^1^	30 ± 3 ^1^	1.35 ± 0.09 ^1^	6.18 ± 0.4 ^1^
Aurora	0.6	45	boiling	219 ± 35 ^1^	300 ± 26 ^1^	50 ± 9	15 ± 1 ^1^	125 ± 5 ^1^	99 ± 4 ^1^	90 ± 14 ^1^	40 ± 4 ^1^	1.78 ± 0.12 ^1^	7.47 ± 0.48 ^1^
Aurora	0.7	45	boiling	199 ± 32 ^1^	253 ± 22 ^1^	52 ± 9	21 ± 2 ^1^	117 ± 5 ^1^	66 ± 3 ^1^	59 ± 9 ^1^	8 ± 1 ^1^	1.94 ± 0.13 ^1^	9.63 ± 0.62 ^1^
Aurora	0.3	90	boiling	514 ± 82 ^0^	553 ± 49 ^0^	124 ± 22 ^0^	19 ± 1 ^0^	216 ± 9 ^0^	423 ± 17 ^0^	284 ± 44 ^0^	39 ± 4 ^0^	0.48 ± 0.03 ^0^	1.42 ± 0.09 ^0^
Aurora	0.4	90	boiling	314 ± 50 ^1^	293 ± 26 ^1^	57 ± 10	24 ± 2 ^1^	134 ± 6 ^1^	369 ± 15 ^1^	275 ± 43 ^0^	47 ± 5 ^1^	0.85 ± 0.06 ^1^	2.79 ± 0.18 ^1^
Aurora	0.5	90	boiling	153 ± 24 ^1^	233 ± 21 ^1^	46 ± 8	12 ± 1 ^1^	103 ± 4 ^1^	62 ± 3 ^1^	55 ± 9 ^1^	4 ± 0 ^1^	1.64 ± 0.11 ^1^	7.01 ± 0.45 ^1^
Aurora	0.6	90	boiling	137 ± 22 ^1^	245 ± 22 ^1^	49 ± 9	16 ± 1 ^1^	122 ± 5 ^1^	54 ± 2 ^1^	54 ± 9 ^1^	11 ± 1 ^1^	1.37 ± 0.09 ^1^	7.59 ± 0.49 ^1^
Aurora	0.7	90	boiling	124 ± 20 ^1^	169 ± 15 ^1^	37 ± 7	20 ± 2 ^1^	100 ± 4 ^1^	63 ± 3 ^1^	56 ± 9 ^1^	24 ± 3 ^1^	2.05 ± 0.13 ^1^	9.20 ± 0.60 ^1^
Aurora	0.3	5	fermentation	257 ± 41 ^0^	804 ± 71 ^0^	385 ± 68 ^0^	67 ± 5 ^0^	101 ± 4 ^0^	157 ± 6 ^0^	126 ± 20 ^0^	95 ± 10 ^0^	0.21 ± 0.01 ^0^	0.31 ± 0.02 ^0^
Aurora	0.4	5	fermentation	151 ± 24 ^1^	501 ± 44 ^1^	478 ± 84 ^0^	39 ± 3 ^1^	63 ± 3 ^1^	170 ± 7 ^1^	147 ± 23 ^0^	105 ± 11 ^0^	0.34 ± 0.02 ^1^	0.63 ± 0.04 ^1^
Aurora	0.5	5	fermentation	130 ± 21 ^1^	393 ± 35 ^1^	386 ± 68 ^0^	39 ± 3 ^1^	45 ± 2 ^1^	74 ± 3 ^1^	65 ± 10 ^1^	41 ± 4 ^1^	0.37 ± 0.02 ^1^	0.92 ± 0.06 ^1^
Aurora	0.6	5	fermentation	121 ± 19 ^1^	432 ± 38 ^1^	296 ± 52	36 ± 3 ^1^	46 ± 2 ^1^	48 ± 2 ^1^	51 ± 8 ^1^	14 ± 2 ^1^	0.53 ± 0.03 ^1^	1.70 ± 0.11 ^1^
Aurora	0.7	5	fermentation	121 ± 19 ^1^	357 ± 31 ^1^	257 ± 45	37 ± 3 ^1^	40 ± 2 ^1^	34 ± 1 ^1^	30 ± 5 ^1^	26 ± 3 ^1^	0.49 ± 0.03 ^1^	1.72 ± 0.11 ^1^
Aurora	0.3	45	fermentation	154 ± 25 ^0^	482 ± 42 ^0^	478 ± 84 ^0^	44 ± 3 ^0^	69 ± 3 ^0^	249 ± 10 ^0^	214 ± 34 ^0^	123 ± 13 ^0^	0.28 ± 0.02 ^0^	0.42 ± 0.03 ^0^
Aurora	0.4	45	fermentation	111 ± 18 ^1^	354 ± 31 ^1^	490 ± 86 ^0^	37 ± 3 ^1^	54 ± 2 ^1^	178 ± 7 ^1^	172 ± 27 ^1^	45 ± 5 ^1^	0.27 ± 0.02 ^0^	0.76 ± 0.05 ^1^
Aurora	0.5	45	fermentation	73 ± 12 ^1^	212 ± 19 ^1^	340 ± 60	34 ± 3 ^1^	39 ± 2 ^1^	46 ± 2 ^1^	43 ± 7 ^1^	21 ± 2 ^1^	0.38 ± 0.02 ^1^	1.32 ± 0.09 ^1^
Aurora	0.6	45	fermentation	71 ± 11 ^1^	206 ± 18 ^1^	309 ± 54 ^1^	30 ± 2 ^1^	35 ± 2 ^1^	35 ± 1 ^1^	37 ± 6 ^1^	1 ± 0 ^1^	0.50 ± 0.03 ^1^	1.74 ± 0.11 ^1^
Aurora	0.7	45	fermentation	59 ± 9 ^1^	177 ± 16 ^1^	305 ± 54 ^1^	28 ± 2 ^1^	36 ± 2 ^1^	26 ± 1 ^1^	28 ± 4 ^1^	30 ± 3 ^1^	0.49 ± 0.03 ^1^	2.35 ± 0.15 ^1^
Aurora	0.3	90	fermentation	138 ± 22 ^0^	508 ± 45 ^0^	402 ± 71 ^0^	44 ± 3 ^0^	66 ± 3 ^0^	109 ± 4 ^0^	96 ± 15 ^0^	132 ± 14 ^0^	0.39 ± 0.03 ^0^	0.44 ± 0.03 ^0^
Aurora	0.4	90	fermentation	81 ± 13 ^1^	258 ± 23 ^1^	425 ± 75 ^0^	33 ± 3 ^1^	50 ± 2 ^1^	138 ± 6 ^1^	133 ± 21 ^1^	47 ± 5 ^1^	0.37 ± 0.02 ^1^	0.95 ± 0.06 ^1^
Aurora	0.5	90	fermentation	61 ± 10 ^1^	136 ± 12 ^1^	337 ± 59 ^1^	27 ± 2 ^1^	37 ± 2 ^1^	29 ± 1 ^1^	27 ± 4 ^1^	40 ± 4 ^1^	0.35 ± 0.02 ^1^	1.20 ± 0.08 ^1^
Aurora	0.6	90	fermentation	63 ± 10 ^1^	194 ± 17 ^1^	325 ± 57 ^1^	29 ± 2 ^1^	35 ± 2 ^1^	25 ± 1 ^1^	22 ± 4 ^1^	44 ± 5 ^1^	0.34 ± 0.02 ^1^	1.52 ± 0.10 ^1^
Aurora	0.7	90	fermentation	73 ± 12 ^1^	219 ± 19 ^1^	328 ± 58 ^1^	51 ± 4 ^1^	34 ± 1 ^1^	18 ± 1 ^1^	20 ± 3 ^1^	41 ± 4 ^1^	0.53 ± 0.03 ^1^	2.09 ± 0.14 ^1^
Aurora	0.3	5	maturation	71 ± 11 ^0^	1032 ± 91 ^0^	327 ± 58 ^0^	81 ± 6 ^0^	140 ± 6 ^0^	31 ± 1 ^0^	23 ± 4 ^0^	94 ± 10 ^0^	0.31 ± 0.02 ^0^	0.37 ± 0.02 ^0^
Aurora	0.4	5	maturation	52 ± 8 ^1^	659 ± 58 ^1^	415 ± 73 ^0^	67 ± 5 ^1^	86 ± 4 ^1^	33 ± 1 ^0^	22 ± 3 ^0^	22 ± 2 ^1^	0.41 ± 0.03 ^1^	0.58 ± 0.04 ^1^
Aurora	0.5	5	maturation	35 ± 6 ^1^	560 ± 49 ^1^	276 ± 49 ^1^	60 ± 5 ^1^	77 ± 3 ^1^	33 ± 1 ^0^	24 ± 4 ^0^	23 ± 2 ^1^	0.55 ± 0.04 ^1^	1.4 ± 0.09 ^1^
Aurora	0.6	5	maturation	45 ± 7 ^1^	675 ± 59 ^1^	285 ± 50 ^1^	56 ± 4 ^1^	77 ± 3 ^1^	48 ± 2 ^1^	42 ± 7 ^1^	24 ± 3 ^1^	0.77 ± 0.05 ^1^	2.14 ± 0.14 ^1^
Aurora	0.7	5	maturation	41 ± 7 ^1^	656 ± 58 ^1^	208 ± 37 ^1^	56 ± 4 ^1^	70 ± 3 ^1^	28 ± 1 ^1^	23 ± 4 ^1^	2 ± 0 ^1^	0.96 ± 0.06 ^1^	3.28 ± 0.21 ^1^
Aurora	0.3	45	maturation	63 ± 10 ^0^	667 ± 59 ^0^	419 ± 74 ^0^	53 ± 4 ^0^	105 ± 5 ^0^	144 ± 6 ^0^	117 ± 18 ^0^	119 ± 13 ^0^	0.31 ± 0.02 ^0^	0.35 ± 0.02 ^0^
Aurora	0.4	45	maturation	38 ± 6 ^1^	494 ± 44 ^1^	415 ± 73 ^0^	52 ± 4 ^0^	80 ± 3 ^1^	103 ± 4 ^1^	86 ± 13 ^1^	19 ± 2 ^1^	0.53 ± 0.03 ^1^	1.15 ± 0.07 ^1^
Aurora	0.5	45	maturation	25 ± 4 ^1^	290 ± 26 ^1^	302 ± 53 ^1^	56 ± 4 ^1^	58 ± 2 ^1^	30 ± 1 ^1^	19 ± 3 ^1^	3 ± 0 ^1^	0.66 ± 0.04 ^1^	2.2 ± 0.14 ^1^
Aurora	0.6	45	maturation	27 ± 4 ^1^	328 ± 29 ^1^	235 ± 42 ^1^	50 ± 4 ^1^	58 ± 3 ^1^	15 ± 1 ^1^	12 ± 2 ^1^	1 ± 0 ^1^	1.07 ± 0.07 ^1^	3.4 ± 0.22 ^1^
Aurora	0.7	45	maturation	25 ± 4 ^1^	294 ± 26 ^1^	307 ± 54 ^1^	67 ± 5 ^1^	56 ± 2 ^1^	16 ± 1 ^1^	14 ± 2 ^1^	1 ± 0 ^1^	0.90 ± 0.06 ^1^	3.87 ± 0.25 ^1^
Aurora	0.3	90	maturation	60 ± 10 ^0^	665 ± 59 ^0^	386 ± 68 ^0^	52 ± 4 ^0^	89 ± 4 ^0^	66 ± 3 ^0^	55 ± 9 ^0^	143 ± 15 ^0^	0.36 ± 0.02 ^0^	0.41 ± 0.03 ^0^
Aurora	0.4	90	maturation	29 ± 5 ^1^	364 ± 32 ^1^	351 ± 62 ^1^	67 ± 5 ^1^	70 ± 3 ^1^	79 ± 3 ^1^	68 ± 11 ^1^	21 ± 2 ^1^	0.65 ± 0.04 ^1^	1.55 ± 0.10 ^1^
Aurora	0.5	90	maturation	< LOD ^1^	193 ± 17 ^1^	259 ± 46 ^1^	51 ± 4 ^1^	56 ± 2 ^1^	17 ± 1 ^1^	13 ± 2 ^1^	46 ± 5 ^1^	0.68 ± 0.04 ^1^	2.44 ± 0.16 ^1^
Aurora	0.6	90	maturation	24 ± 4 ^1^	279 ± 25 ^1^	273 ± 48 ^1^	50 ± 4 ^1^	58 ± 2 ^1^	13 ± 1 ^1^	9 ± 1 ^1^	88 ± 9 ^1^	0.50 ± 0.03 ^1^	2.48 ± 0.16 ^1^
Aurora	0.7	90	maturation	27 ± 4 ^1^	276 ± 24 ^1^	260 ± 46 ^1^	44 ± 3 ^1^	55 ± 2 ^1^	9 ± 0 ^1^	6 ± 1 ^1^	16 ± 2 ^1^	0.98 ± 0.06 ^1^	3.70 ± 0.24 ^1^

The analysis of variance (ANOVA) was carried out at 0.05 significance level in order to differentiate between samples brewed with fresh hops with HSI 0.3 (0) and other samples with significant differences (1).

**Table 5 foods-12-04353-t005:** Content of hop aroma compounds in beer samples brewed with Styrian Wolf variety determined by HS-SPME-GC-MS.

Variety	HSI	Boiling Time	Sampling	Myrcene [μg/L]	Linalool [μg/L]	Alpha-Terpineol [μg/L]	Beta-Citronellol [μg/L]	Geraniol [μg/L]	Beta-Caryophyllene [μg/L]	Alpha-Humulene [μg/L]	Caryophyllene Oxide [μg/L]	Humulene Epoxide I [rel. %]	Humulenol II [rel. %]
Styrian Wolf	0.3	5	boiling	3508 ± 561 ^0^	782 ± 69 ^0^	48 ± 8 ^0^	30 ± 2 ^0^	536 ± 23 ^0^	427 ± 17 ^0^	236 ± 37 ^0^	33 ± 4 ^0^	0.3 ± 0.02 ^0^	1.18 ± 0.08 ^0^
Styrian Wolf	0.4	5	boiling	1461 ± 234 ^1^	845 ± 74 ^0^	54 ± 10 ^0^	29 ± 2 ^0^	452 ± 19 ^1^	286 ± 12 ^1^	167 ± 26 ^0^	61 ± 7 ^1^	0.48 ± 0.03 ^1^	1.89 ± 0.12 ^1^
Styrian Wolf	0.5	5	boiling	1347 ± 215 ^1^	806 ± 71 ^0^	55 ± 10 ^0^	32 ± 2 ^0^	480 ± 21 ^1^	282 ± 11 ^1^	165 ± 26 ^0^	50 ± 5 ^1^	0.57 ± 0.04 ^1^	2.26 ± 0.15 ^1^
Styrian Wolf	0.6	5	boiling	1202 ± 192 ^1^	778 ± 69 ^0^	55 ± 10 ^0^	29 ± 2 ^0^	417 ± 18 ^1^	176 ± 7 ^1^	112 ± 18 ^1^	53 ± 6 ^1^	0.76 ± 0.05 ^1^	3.05 ± 0.2 ^1^
Styrian Wolf	0.7	5	boiling	1081 ± 173 ^1^	967 ± 85 ^0^	72 ± 13 ^0^	38 ± 3 ^0^	531 ± 23 ^1^	186 ± 7 ^1^	124 ± 19 ^1^	54 ± 6 ^1^	0.8 ± 0.05 ^1^	3.52 ± 0.23 ^1^
Styrian Wolf	0.3	45	boiling	1274 ± 204 ^0^	755 ± 67 ^0^	58 ± 10 ^0^	31 ± 2 ^0^	548 ± 24 ^0^	479 ± 19 ^0^	261 ± 41 ^0^	45 ± 5 ^0^	0.35 ± 0.02 ^0^	1.16 ± 0.08 ^0^
Styrian Wolf	0.4	45	boiling	389 ± 62 ^1^	362 ± 32 ^1^	38 ± 7 ^1^	24 ± 2 ^1^	404 ± 17 ^1^	266 ± 11 ^1^	175 ± 27 ^1^	79 ± 8 ^1^	0.87 ± 0.06 ^1^	2.82 ± 0.18 ^1^
Styrian Wolf	0.5	45	boiling	421 ± 67 ^1^	475 ± 42 ^1^	47 ± 8 ^1^	27 ± 2 ^1^	428 ± 18 ^1^	305 ± 12 ^1^	179 ± 28 ^1^	23 ± 3 ^1^	0.69 ± 0.04 ^1^	2.52 ± 0.16 ^1^
Styrian Wolf	0.6	45	boiling	401 ± 64 ^1^	398 ± 35 ^1^	45 ± 8 ^1^	24 ± 2 ^1^	322 ± 14 ^1^	223 ± 9 ^1^	100 ± 16 ^1^	29 ± 3 ^1^	1.60 ± 0.10 ^1^	5.79 ± 0.38 ^1^
Styrian Wolf	0.7	45	boiling	316 ± 51 ^1^	319 ± 28 ^1^	44 ± 8 ^1^	20 ± 1 ^1^	293 ± 13 ^1^	108 ± 4 ^1^	67 ± 10 ^1^	32 ± 3 ^1^	1.06 ± 0.07 ^1^	4.90 ± 0.32 ^1^
Styrian Wolf	0.3	90	boiling	970 ± 155 ^0^	803 ± 71 ^0^	73 ± 13 ^0^	38 ± 3 ^0^	635 ± 27 ^0^	377 ± 15 ^0^	222 ± 35 ^0^	72 ± 8 ^0^	0.58 ± 0.04 ^0^	1.59 ± 0.10 ^0^
Styrian Wolf	0.4	90	boiling	324 ± 52 ^1^	479 ± 42 ^1^	50 ± 9 ^1^	30 ± 2 ^1^	506 ± 22 ^1^	251 ± 10 ^1^	143 ± 22 ^1^	61 ± 6 ^1^	0.59 ± 0.04 ^0^	1.89 ± 0.12 ^1^
Styrian Wolf	0.5	90	boiling	321 ± 51 ^1^	498 ± 44 ^1^	52 ± 9 ^1^	27 ± 2 ^1^	405 ± 17 ^1^	285 ± 12 ^1^	165 ± 26 ^1^	45 ± 5 ^1^	0.62 ± 0.04 ^1^	1.94 ± 0.13 ^1^
Styrian Wolf	0.6	90	boiling	216 ± 35 ^1^	323 ± 28 ^1^	47 ± 8 ^1^	23 ± 2 ^1^	326 ± 14 ^1^	68 ± 3 ^1^	50 ± 8 ^1^	12 ± 1 ^1^	1.06 ± 0.07 ^1^	4.02 ± 0.26 ^1^
Styrian Wolf	0.7	90	boiling	229 ± 37 ^1^	376 ± 33 ^1^	52 ± 9 ^1^	26 ± 2 ^1^	353 ± 15 ^1^	134 ± 5 ^1^	92 ± 14 ^1^	9 ± 1 ^1^	0.99 ± 0.06 ^1^	3.65 ± 0.24 ^1^
Styrian Wolf	0.3	5	fermentation	349 ± 56 ^0^	599 ± 53 ^0^	391 ± 69 ^0^	116 ± 9 ^0^	152 ± 7 ^0^	132 ± 5 ^0^	93 ± 15 ^0^	1 ± 1 ^0^	0.15 ± 0.01 ^0^	0.31 ± 0.02 ^0^
Styrian Wolf	0.4	5	fermentation	279 ± 45 ^0^	688 ± 61 ^0^	378 ± 67 ^0^	133 ± 10 ^0^	154 ± 7 ^0^	112 ± 5 ^1^	80 ± 13 ^0^	14 ± 2 ^1^	0.23 ± 0.01 ^1^	0.51 ± 0.03 ^1^
Styrian Wolf	0.5	5	fermentation	248 ± 40 ^1^	504 ± 44 ^1^	358 ± 63 ^0^	151 ± 12 ^1^	121 ± 5 ^1^	119 ± 5 ^1^	96 ± 15 ^0^	5 ± 1 ^1^	0.33 ± 0.02 ^1^	1.01 ± 0.07 ^1^
Styrian Wolf	0.6	5	fermentation	238 ± 38 ^1^	536 ± 47 ^1^	366 ± 65 ^0^	104 ± 8 ^1^	110 ± 5 ^1^	81 ± 3 ^1^	63 ± 10 ^1^	4 ± 0 ^1^	0.32 ± 0.02 ^1^	0.89 ± 0.06 ^1^
Styrian Wolf	0.7	5	fermentation	239 ± 38 ^1^	641 ± 56 ^1^	342 ± 60 ^0^	121 ± 9 ^1^	106 ± 5 ^1^	60 ± 2 ^1^	47 ± 7 ^1^	2 ± 1 ^1^	0.34 ± 0.02 ^1^	1.11 ± 0.07 ^1^
Styrian Wolf	0.3	45	fermentation	298 ± 48 ^0^	607 ± 53 ^0^	405 ± 72 ^0^	132 ± 10 ^0^	151 ± 7 ^0^	146 ± 6 ^0^	103 ± 16 ^0^	38 ± 4 ^0^	0.19 ± 0.01 ^0^	0.42 ± 0.03 ^0^
Styrian Wolf	0.4	45	fermentation	128 ± 20 ^1^	235 ± 21 ^1^	374 ± 66 ^1^	78 ± 6 ^1^	90 ± 4 ^1^	97 ± 4 ^1^	82 ± 13 ^1^	4 ± 1 ^1^	0.35 ± 0.02 ^1^	0.83 ± 0.05 ^1^
Styrian Wolf	0.5	45	fermentation	168 ± 27 ^1^	319 ± 28 ^1^	364 ± 64 ^1^	95 ± 7 ^1^	13 ± 1 ^1^	103 ± 4 ^1^	85 ± 13 ^1^	4 ± 1 ^1^	0.28 ± 0.02 ^1^	0.74 ± 0.05 ^1^
Styrian Wolf	0.6	45	fermentation	131 ± 21 ^1^	250 ± 22 ^1^	374 ± 66	78 ± 6 ^1^	82 ± 4 ^1^	39 ± 2 ^1^	31 ± 5 ^1^	6 ± 1 ^1^	0.38 ± 0.02 ^1^	1.33 ± 0.09 ^1^
Styrian Wolf	0.7	45	fermentation	127 ± 20 ^1^	246 ± 22 ^1^	403 ± 71	70 ± 5 ^1^	13 ± 1 ^1^	39 ± 2 ^1^	29 ± 5 ^1^	6 ± 1 ^1^	0.33 ± 0.02 ^1^	1.49 ± 0.10 ^1^
Styrian Wolf	0.3	90	fermentation	297 ± 48 ^0^	665 ± 59 ^0^	448 ± 79 ^0^	133 ± 10 ^0^	150 ± 6 ^0^	136 ± 5 ^0^	103 ± 16 ^0^	40 ± 4 ^0^	0.22 ± 0.01 ^0^	0.56 ± 0.04 ^0^
Styrian Wolf	0.4	90	fermentation	193 ± 31 ^1^	472 ± 42 ^1^	465 ± 82 ^0^	115 ± 9 ^1^	119 ± 5 ^1^	106 ± 4 ^1^	84 ± 13 ^1^	5 ± 1 ^1^	0.27 ± 0.02 ^1^	0.87 ± 0.06 ^1^
Styrian Wolf	0.5	90	fermentation	275 ± 44 ^1^	581 ± 51 ^1^	360 ± 64 ^1^	118 ± 9 ^1^	135 ± 6 ^1^	92 ± 4 ^1^	63 ± 10 ^1^	1 ± 1 ^1^	0.22 ± 0.01 ^1^	0.48 ± 0.03 ^1^
Styrian Wolf	0.6	90	fermentation	149 ± 24 ^1^	279 ± 25 ^1^	364 ± 64 ^1^	82 ± 6 ^1^	85 ± 4 ^1^	27 ± 1 ^1^	23 ± 4 ^1^	8 ± 1 ^1^	0.36 ± 0.02 ^1^	1.33 ± 0.09 ^1^
Styrian Wolf	0.7	90	fermentation	154 ± 25 ^1^	289 ± 25 ^1^	348 ± 61 ^1^	96 ± 7 ^1^	86 ± 4 ^1^	45 ± 2 ^1^	42 ± 7 ^1^	9 ± 1 ^1^	0.44 ± 0.03 ^1^	1.54 ± 0.10 ^1^
Styrian Wolf	0.3	5	maturation	163 ± 26 ^0^	917 ± 81 ^0^	393 ± 69 ^0^	147 ± 11 ^0^	269 ± 12 ^0^	66 ± 3 ^0^	45 ± 7 ^0^	14 ± 1 ^0^	0.22 ± 0.01 ^0^	0.38 ± 0.02 ^0^
Styrian Wolf	0.4	5	maturation	134 ± 21 ^0^	930 ± 82 ^0^	344 ± 61 ^0^	138 ± 11 ^0^	230 ± 10 ^1^	53 ± 2 ^1^	37 ± 6 ^0^	23 ± 2 ^1^	0.32 ± 0.02 ^1^	0.68 ± 0.04 ^1^
Styrian Wolf	0.5	5	maturation	90 ± 14 ^1^	670 ± 59 ^1^	318 ± 56 ^0^	216 ± 16 ^1^	186 ± 8 ^1^	41 ± 2 ^1^	29 ± 5 ^1^	8 ± 1 ^1^	0.61 ± 0.04 ^1^	1.94 ± 0.13 ^1^
Styrian Wolf	0.6	5	maturation	104 ± 17 ^1^	720 ± 63 ^1^	257 ± 45 ^1^	106 ± 8 ^1^	172 ± 7 ^1^	29 ± 1 ^1^	20 ± 3 ^1^	14 ± 1 ^1^	0.51 ± 0.03 ^1^	1.43 ± 0.09 ^1^
Styrian Wolf	0.7	5	maturation	117 ± 19 ^1^	850 ± 75 ^1^	327 ± 58 ^1^	128 ± 10 ^1^	200 ± 9 ^1^	22 ± 1 ^1^	16 ± 3 ^1^	6 ± 1 ^1^	0.54 ± 0.03 ^1^	1.67 ± 0.11 ^1^
Styrian Wolf	0.3	45	maturation	133 ± 21 ^0^	829 ± 73 ^0^	353 ± 62 ^0^	155 ± 12 ^0^	245 ± 11 ^0^	104 ± 4 ^0^	71 ± 11 ^0^	2 ± 1 ^0^	0.24 ± 0.02 ^0^	0.43 ± 0.03 ^0^
Styrian Wolf	0.4	45	maturation	73 ± 12 ^1^	379 ± 33 ^1^	352 ± 62 ^0^	118 ± 9 ^1^	155 ± 7 ^1^	49 ± 2 ^1^	38 ± 6 ^1^	6 ± 1 ^1^	0.54 ± 0.03 ^1^	1.44 ± 0.09 ^1^
Styrian Wolf	0.5	45	maturation	83 ± 13 ^1^	510 ± 45 ^1^	325 ± 57 ^0^	139 ± 11 ^1^	160 ± 7 ^1^	63 ± 3 ^1^	46 ± 7 ^1^	4 ± 1 ^1^	0.5 ± 0.03 ^1^	1.54 ± 0.10 ^1^
Styrian Wolf	0.6	45	maturation	64 ± 10 ^1^	360 ± 32 ^1^	295 ± 52 ^1^	105 ± 8 ^1^	139 ± 6 ^1^	14 ± 1 ^1^	9 ± 1 ^1^	2 ± 1 ^1^	0.54 ± 0.03 ^1^	1.95 ± 0.13 ^1^
Styrian Wolf	0.7	45	maturation	66 ± 11 ^1^	326 ± 29 ^1^	342 ± 60 ^1^	103 ± 8 ^1^	126 ± 5 ^1^	16 ± 1 ^1^	10 ± 2 ^1^	1 ± 1 ^1^	0.55 ± 0.04 ^1^	2.14 ± 0.14 ^1^
Styrian Wolf	0.3	90	maturation	166 ± 27 ^0^	933 ± 82 ^0^	449 ± 79 ^0^	162 ± 12 ^0^	244 ± 11 ^0^	661 ± 27 ^0^	44 ± 7 ^0^	74 ± 8 ^0^	0.41 ± 0.03 ^0^	0.59 ± 0.04 ^0^
Styrian Wolf	0.4	90	maturation	101 ± 16 ^1^	566 ± 50 ^1^	353 ± 62 ^1^	128 ± 10 ^1^	175 ± 8 ^1^	32 ± 1 ^1^	23 ± 4 ^1^	50 ± 5 ^1^	0.41 ± 0.03 ^1^	1.37 ± 0.09 ^1^
Styrian Wolf	0.5	90	maturation	89 ± 14 ^1^	767 ± 68 ^1^	290 ± 51 ^1^	153 ± 12 ^1^	217 ± 9 ^1^	37 ± 1 ^1^	26 ± 4 ^1^	20 ± 2 ^1^	0.34 ± 0.02 ^1^	0.75 ± 0.05 ^1^
Styrian Wolf	0.6	90	maturation	75 ± 12 ^1^	334 ± 29 ^1^	294 ± 52 ^1^	129 ± 10 ^1^	119 ± 5 ^1^	14 ± 1 ^1^	7 ± 1 ^1^	3 ± 1 ^1^	0.68 ± 0.04 ^1^	2.45 ± 0.16 ^1^
Styrian Wolf	0.7	90	maturation	79 ± 13 ^1^	405 ± 36 ^1^	303 ± 54 ^1^	147 ± 11 ^1^	149 ± 6 ^1^	19 ± 1 ^1^	14 ± 2 ^1^	5 ± 1 ^1^	0.68 ± 0.04 ^1^	2.68 ± 0.17 ^1^

The analysis of variance (ANOVA) was carried out at 0.05 significance level in order to differentiate between samples brewed with fresh hops with HSI 0.3 (0) and other samples with significant differences (1).

**Table 6 foods-12-04353-t006:** Results of sensorial analysis for quality and intensity of aroma.

Variety	HSI	Boiling Time [min]	Quality of Aroma	Intensity of Aroma
Celeia	0.3	5	4.7 ± 0.3	3.6 ± 0.3
Celeia	0.4	5	4.2 ± 0.3	2.9 ± 0.4
Celeia	0.5	5	3.0 ± 0.3	2.2 ± 0.4
Celeia	0.6	5	1.3 ± 0.4	1.6 ± 0.3
Celeia	0.7	5	2.0 ± 0.2	1.5 ± 0.4
Celeia	0.3	45	4.9 ± 0.2	2.7 ± 0.3
Celeia	0.4	45	4.9 ± 0.3	2.2 ± 0.3
Celeia	0.5	45	4.3 ± 0.3	1.6 ± 0.3
Celeia	0.6	45	3.4 ± 0.2	0.8 ± 0.3
Celeia	0.7	45	3.1 ± 0.4	0.2 ± 0.3
Celeia	0.3	90	4.2 ± 0.3	1.4 ± 0.3
Celeia	0.4	90	2.9 ± 0.3	0.8 ± 0.2
Celeia	0.5	90	3.0 ± 0.2	0.9 ± 0.2
Celeia	0.6	90	3.2 ± 0.4	0.7 ± 0.3
Celeia	0.7	90	2.8 ± 0.5	0.8 ± 0.2
Aurora	0.3	5	4.9 ± 0.2	3.2 ± 0.3
Aurora	0.4	5	4.4 ± 0.2	2.4 ± 0.2
Aurora	0.5	5	4.0 ± 0.2	1.7 ± 0.3
Aurora	0.6	5	3.1 ± 0.2	1.1 ± 0.2
Aurora	0.7	5	3.1 ± 0.2	1.1 ± 0.2
Aurora	0.3	45	4.0 ± 0.2	2.9 ± 0.2
Aurora	0.4	45	4.0 ± 0.4	2.5 ± 0.2
Aurora	0.5	45	3.1 ± 0.3	1.8 ± 0.3
Aurora	0.6	45	0.2 ± 0.3	1.8 ± 0.2
Aurora	0.7	45	0.3 ± 0.3	1.8 ± 0.3
Aurora	0.3	90	1.5 ± 0.3	2.1 ± 0.2
Aurora	0.4	90	0.4 ± 0.2	1.2 ± 0.3
Aurora	0.5	90	0.3 ± 0.3	1.3 ± 0.2
Aurora	0.6	90	0.2 ± 0.3	1.2 ± 0.3
Aurora	0.7	90	0.3 ± 0.3	1.3 ± 0.2
Styrian Wolf	0.3	5	4.9 ± 0.2	3.6 ± 0.3
Styrian Wolf	0.4	5	4.7 ± 0.3	3.6 ± 0.4
Styrian Wolf	0.5	5	4.0 ± 0.2	2.9 ± 0.3
Styrian Wolf	0.6	5	3.4 ± 0.3	2.2 ± 0.2
Styrian Wolf	0.7	5	3.6 ± 0.4	1.4 ± 0.2
Styrian Wolf	0.3	45	4.8 ± 0.5	2.9 ± 0.2
Styrian Wolf	0.4	45	4.8 ± 0.4	2.5 ± 0.2
Styrian Wolf	0.5	45	3.9 ± 0.2	2.0 ± 0.0
Styrian Wolf	0.6	45	3.2 ± 0.4	1.5 ± 0.2
Styrian Wolf	0.7	45	2.7 ± 0.2	1.4 ± 0.2
Styrian Wolf	0.3	90	4.8 ± 0.3	2.6 ± 0.2
Styrian Wolf	0.4	90	4.5 ± 0.3	2.1 ± 0.3
Styrian Wolf	0.5	90	3.4 ± 0.6	1.7 ± 0.2
Styrian Wolf	0.6	90	3.0 ± 0.2	1.2 ± 0.2
Styrian Wolf	0.7	90	2.4 ± 0.2	1.1 ± 0.3

**Table 7 foods-12-04353-t007:** Results of chemical analysis for BU, alpha-acids, humulinones and iso-alpha-acids by UV-VIS (BU) and HPLC (alpha-acids, humulinones, iso-alpha-acids).

Variety	HSI	Boiling Time [min]	Bitterness [BU]	Alpha-Acids [mg/L]	Humulinones [mg/L]	Iso-Alpha-Acids [mg/L]
Celeia	0.3	5	45 ± 2	8.23 ± 0.53	8.70 ± 0.25	18.03 ± 0.35
Celeia	0.4	5	28 ± 1	3.83 ± 0.25	11.83 ± 0.35	12.36 ± 0.24
Celeia	0.5	5	32 ± 1	4.00 ± 0.26	14.58 ± 0.43	11.09 ± 0.21
Celeia	0.6	5	33 ± 1	4.09 ± 0.26	14.96 ± 0.44	11.43 ± 0.22
Celeia	0.7	5	37 ± 2	2.76 ± 0.18	14.77 ± 0.43	10.03 ± 0.19
Celeia	0.3	45	41 ± 2	2.94 ± 0.19	6.98 ± 0.20	61.13 ± 1.18
Celeia	0.4	45	41 ± 2	<LOD *	9.96 ± 0.29	43.94 ± 0.85
Celeia	0.5	45	51 ± 2	<LOD *	14.04 ± 0.41	50.17 ± 0.97
Celeia	0.6	45	44 ± 2	<LOD *	14.16 ± 0.41	40.6 ± 0.78
Celeia	0.7	45	47 ± 2	<LOD *	14.04 ± 0.40	39.96 ± 0.77
Celeia	0.3	90	56 ± 2	<LOD *	5.00 ± 0.15	81.28 ± 1.57
Celeia	0.4	90	58 ± 2	<LOD *	11.03 ± 0.32	64.35 ± 1.24
Celeia	0.5	90	51 ± 2	<LOD *	11.18 ± 0.33	52.62 ± 1.02
Celeia	0.6	90	49 ± 2	<LOD *	11.80 ± 0.34	49.95 ± 0.96
Celeia	0.7	90	53 ± 2	<LOD *	12.44 ± 0.36	46.36 ± 0.89
Aurora	0.3	5	46 ± 2	7.07 ± 0.46	19.00 ± 0.55	37.59 ± 0.73
Aurora	0.4	5	71 ± 3	8.63 ± 0.56	33.50 ± 0.98	40.3 ± 0.78
Aurora	0.5	5	81 ± 3	6.98 ± 0.45	47.69 ± 1.39	23.58 ± 0.46
Aurora	0.6	5	80 ± 3	8.81 ± 0.57	42.66 ± 1.25	17.65 ± 0.34
Aurora	0.7	5	84 ± 4	5.78 ± 0.37	44.11 ± 1.29	13.55 ± 0.26
Aurora	0.3	45	105 ± 4	4.09 ± 0.26	17.28 ± 0.50	144.02 ± 2.78
Aurora	0.4	45	115 ± 5	2.94 ± 0.19	29.72 ± 0.87	140.42 ± 2.71
Aurora	0.5	45	116 ± 5	2.40 ± 0.16	41.55 ± 1.21	103.76 ± 2
Aurora	0.6	45	107 ± 4	<LOD *	35.52 ± 1.04	88.35 ± 1.71
Aurora	0.7	45	122 ± 5	<LOD *	40.75 ± 1.19	85.43 ± 1.65
Aurora	0.3	90	116 ± 5	1.56 ± 0.11	16.18 ± 0.47	174.29 ± 3.36
Aurora	0.4	90	117 ± 5	<LOD *	24.38 ± 0.71	145.84 ± 2.81
Aurora	0.5	90	120 ± 5	<LOD *	34.74 ± 1.01	119.30 ± 2.30
Aurora	0.6	90	116 ± 5	<LOD *	33.12 ± 0.97	105.03 ± 2.03
Aurora	0.7	90	112 ± 5	<LOD *	31.33 ± 0.91	88.95 ± 1.72
Styrian Wolf	0.3	5	56 ± 2	7.16 ± 0.46	24.00 ± 0.70	32.05 ± 0.62
Styrian Wolf	0.4	5	59 ± 2	5.65 ± 0.37	38.01 ± 1.11	22.49 ± 0.43
Styrian Wolf	0.5	5	76 ± 3	6.63 ± 0.43	51.51 ± 1.5	16.68 ± 0.32
Styrian Wolf	0.6	5	52 ± 2	6.23 ± 0.41	36.86 ± 1.08	12.40 ± 0.24
Styrian Wolf	0.7	5	63 ± 3	5.60 ± 0.36	37.54 ± 1.10	11.74 ± 0.23
Styrian Wolf	0.3	45	86 ± 4	4 ± 0.26	20.18 ± 0.59	120.44 ± 2.32
Styrian Wolf	0.4	45	93 ± 4	2.8 ± 0.18	35.94 ± 1.05	103.89 ± 2.01
Styrian Wolf	0.5	45	104 ± 4	2.27 ± 0.15	44.53 ± 1.30	85.64 ± 1.65
Styrian Wolf	0.6	45	105 ± 4	2.94 ± 0.19	39.26 ± 1.15	92.04 ± 1.78
Styrian Wolf	0.7	45	97 ± 4	2.45 ± 0.16	39.45 ± 1.15	75.40 ± 1.46
Styrian Wolf	0.3	90	106 ± 4	1.87 ± 0.12	16.6 ± 0.48	148.43 ± 2.86
Styrian Wolf	0.4	90	101 ± 4	1.65 ± 0.11	31.97 ± 0.93	119.60 ± 2.31
Styrian Wolf	0.5	90	118 ± 5	<LOD *	40.56 ± 1.18	108.50 ± 2.09
Styrian Wolf	0.6	90	89 ± 4	<LOD *	32.62 ± 0.95	82.85 ± 1.60
Styrian Wolf	0.7	90	96 ± 4	<LOD *	34.23 ± 1.00	89.45 ± 1.73

* LOD (limit of detection) < 1.34 mg/L.

**Table 8 foods-12-04353-t008:** Results of sensorial analysis for quality and intensity of bitterness.

Variety	HSI	Boiling Time [min]	Quality of Bitterness	Intensity of Bitterness
Celeia	0.3	5	4.8 ± 0.4	1.1 ± 0.3
Celeia	0.4	5	4.4 ± 0.4	1.0 ± 0.2
Celeia	0.5	5	3.1 ± 0.6	1.8 ± 0.4
Celeia	0.6	5	1.5 ± 0.3	2.0 ± 0.2
Celeia	0.7	5	1.0 ± 0.4	1.9 ± 0.2
Celeia	0.3	45	3.0 ± 0.3	2.5 ± 0.2
Celeia	0.4	45	3.1 ± 0.4	2.5 ± 0.2
Celeia	0.5	45	3.0 ± 0.3	2.4 ± 0.2
Celeia	0.6	45	3.9 ± 0.3	1.7 ± 0.2
Celeia	0.7	45	3.8 ± 0.4	1.5 ± 0.2
Celeia	0.3	90	3.1 ± 0.3	3.0 ± 0.2
Celeia	0.4	90	3.1 ± 0.4	3.0 ± 0.2
Celeia	0.5	90	3.0 ± 0.4	2.9 ± 0.2
Celeia	0.6	90	3.7 ± 0.4	2.9 ± 0.2
Celeia	0.7	90	3.8 ± 0.3	2.1 ± 0.2
Styrian Wolf	0.3	5	4.9 ± 0.2	3.1 ± 0.2
Styrian Wolf	0.4	5	4.1 ± 0.2	3.2 ± 0.2
Styrian Wolf	0.5	5	2.1 ± 0.3	4.2 ± 0.4
Styrian Wolf	0.6	5	3.4 ± 0.2	4.3 ± 0.4
Styrian Wolf	0.7	5	3.6 ± 0.2	4.2 ± 0.2
Styrian Wolf	0.3	45	3.0 ± 0.0	5.0 ± 0.2
Styrian Wolf	0.4	45	3.0 ± 0.2	5.0 ± 0.2
Styrian Wolf	0.5	45	2.5 ± 0.2	4.6 ± 0.3
Styrian Wolf	0.6	45	2.4 ± 0.2	4.6 ± 0.4
Styrian Wolf	0.7	45	2.4 ± 0.2	4.4 ± 0.2
Styrian Wolf	0.3	90	2.3 ± 0.3	4.9 ± 0.2
Styrian Wolf	0.4	90	2.1 ± 0.2	4.9 ± 0.2
Styrian Wolf	0.5	90	4.1 ± 0.2	3.4 ± 0.3
Styrian Wolf	0.6	90	3.1 ± 0.2	3.8 ± 0.2
Styrian Wolf	0.7	90	3.1 ± 0.2	3.9 ± 0.2

## Data Availability

Data are contained within the article.

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
