# Peer review of "The Impact of Hop Freshness on Kettle-Hopped Beers"

_foods, 2023, doi:10.3390/foods12234353_

Round 1
Reviewer 1 Report
Comments and Suggestions for Authors
This manuscript is about quality analysis on Hop Fresh on Kettle-Hopped Beers. The authors carried out many experiments. However, there are a few things that need to be revised.
Table organization is inconsistent. Please unify the table border and layout. Additional statistical display is also required for the readability of many data in the table.
The Figures are small and not clear, so it's hard to distinguish.
Check the references to revise them Please match the form for foods journal
Author Response
Reviewer: 1
- Table organization is inconsistent. Please unify the table border and layout. Additional statistical display is also required for the readability of many data in the table.
We unified the appearance of the tables and we added additional statistical display in order to show if there and where is the difference between beers brewed with hops with different HSI, same variety and same boiling time.
- The Figures are small and not clear, so it's hard to distinguish.
We have enlarged the Figures for better visibility.
- Check the references to revise them Please match the form for foods journal
We revised the references. The form for food journal is accepting any kind of style since the following sentence is found on their page: ''Your references may be in any style, provided that you use the consistent formatting throughout. It is essential to include author(s) name(s), journal or book title, article or chapter title (where required), year of publication, volume and issue (where appropriate) and pagination. DOI numbers (Digital Object Identifier) are not mandatory but highly encouraged. ''

Reviewer 2 Report
Comments and Suggestions for Authors
I found the topic of this research article interesting. Still, there are some changes that can be made:
Pay attention to all Latin denominations, they all should be written with italics.
There are some spelling errors in the entire manuscript. Please check it carefully.
The Introduction section is too long and is not comprehensive enough to make the reader understand the aim of this paper. Please restructure it and focus more on the impact of your findings to the industrials. As it is an Open access paper, brewing actors will be able to reach your paper.
Use a figure also to present the experimental variants developed in this study. In this figure insert also all the abbreviations you used in the study. They are currently missing from the beer-obtaining process description.
Is statistical analysis missing? The results from the tables are not expressed correctly. And in many cases, the standard deviations are very high. Explain the reason.
How many replicates were done?
Results and discussion should be improved by adding citing references to support the explanations and your findings. The comparison to previous findings is almost missing.
Comments on the Quality of English LanguageEnglish language style must be improved.
Author Response
Reviewer: 2
- Pay attention to all Latin denominations, they all should be written with italics.
We corrected the style of words Humulus lupulus L.
- There are some spelling errors in the entire manuscript. Please check it carefully.
We have checked for spelling errors and all errors found were corrected.
- The Introduction section is too long and is not comprehensive enough to make the reader understand the aim of this paper. Please restructure it and focus more on the impact of your findings to the industrials. As it is an Open access paper, brewing actors will be able to reach your paper.
We shortened the introduction section and restructured it.
- Use a figure also to present the experimental variants developed in this study. In this figure insert also all the abbreviations you used in the study. They are currently missing from the beer-obtaining process description.
We added the figures which explain the experimental part of brewing trials. We are not sure which abbreviations are you having in mind, since we did not use any of abbreviations in the study, except HSI which have been clarified in abstract and introduction.
- Is statistical analysis missing? The results from the tables are not expressed correctly. And in many cases, the standard deviations are very high. Explain the reason.
Since all measurements were performed in duplicates, that is not enough to express meaningful standard deviation, therefore the SD was calculated from RSD. Concentrations of individual compounds are very different and therefore also the SD values are not comparable. If RDSs were presented the values would be much more equal.
- How many replicates were done?
There were one brewing batch of every sample (altogether 45 samples) and further all measurement were made in duplicate. This is also clearer now, when brewing scheme is added for each variety.
- Results and discussion should be improved by adding citing references to support the explanations and your findings. The comparison to previous findings is almost missing.
The work on this field is relatively new and beside 2 mentioned researches (which focused on other parameters) we did not find any other relevant reference.

Reviewer 3 Report
Comments and Suggestions for Authors
The work is certainly good, but I still have some points to clarify, as they seem not well described.
- To determine the presence of some compounds, the authors used HPLC and HS-SPME techniques. It is not sufficiently clear what was determined with which technique. Since at least a couple of chromatograms are missing as examples, I believe this aspect can be made clearer by modifying the header of the tables, or by adding some descriptive element to the footer, or along the text.
- Lines 110, 111 - … international calibration extract 4 (ICE4), DCHA-Iso and DCHA-Humulinones … what are these materials? The meaning of these acronyms?
- Paragraph 2.3. Beer brewing, and line 134 – If I understand correctly, 45 batches of beer were brewed.
-line 134: All experiment were conducted in duplicate ... starting from the production of beer?, or were only the samples (at boiling, fermentation and maturation step) analyzed in duplicate?
- line 140 – Each panellist tested 15 beer samples (over 45 batches in total ??) obtained from the same hop cultivar. If I understand correctly, each beer sample obtained 3 evaluation sheets. If so, it can be specified further. In any case, I suggest clarifying the paragraph better, indicating the number of evaluation forms for each beer sample.
- line 158 – we need to add … ‘absorbance’….
- lines 219-221 – How was the standard deviation obtained ? how many replicates ?
- Tables 2,3,4,5,6,7 report compositional data with standard deviations. It is unclear how the std dev was obtained, and how many replications were performed.
Author Response
Reviewer: 3
- To determine the presence of some compounds, the authors used HPLC and HS-SPME techniques. It is not sufficiently clear what was determined with which technique. Since at least a couple of chromatograms are missing as examples, I believe this aspect can be made clearer by modifying the header of the tables, or by adding some descriptive element to the footer, or along the text.
We added the information into tables header.
- Lines 110, 111 - … international calibration extract 4 (ICE4), DCHA-Iso and DCHA-Humulinones … what are these materials? The meaning of these acronyms?
These materials are certified reference materials for determination of alpha-acids and beta acids (ICE4), iso-alpha-acids (DCHA-Iso) and humulinones (DCHA-Humulinones) approved by analytical commission ofEBC and they are listed in the official analytical methods according to Analytica-EBC. We also added the explanation into the manuscript.
- Paragraph 2.3. Beer brewing, and line 134 – If I understand correctly, 45 batches of beer were brewed.
Yes, that is correct. We also added the scheme of brewing trial for easier understanding.
- -line 134: All experiment were conducted in duplicate ... starting from the production of beer?, or were only the samples (at boiling, fermentation and maturation step) analyzed in duplicate?
Only samples were analysed in duplicate. We think that with schemes of brewing trials that is clearer. In any case we were following the trends for each particular compound through increasing HSI values.
- line 140 – Each panellist tested 15 beer samples (over 45 batches in total ??) obtained from the same hop cultivar. If I understand correctly, each beer sample obtained 3 evaluation sheets. If so, it can be specified further. In any case, I suggest clarifying the paragraph better, indicating the number of evaluation forms for each beer sample.
We rewrite the whole paragraph for better understanding of procedure of sensorial analysis.
- line 158 – we need to add … ‘absorbance’….
We added the absorbance.
- lines 219-221 – How was the standard deviation obtained ? how many replicates ?
SD was calculated from RSD of methods validation parameters. Samples were measured in duplicates.
- Tables 2,3,4,5,6,7 report compositional data with standard deviations. It is unclear how the std dev was obtained, and how many replications were performed.
Tables reporting results of chemical analysis are presenting mean value measured from duplicates and SD calculated from RSD of validated method. Tables presenting results of sensorial analysis contain mean value of 10 evaluation (10 assessors) and SD was calculated from those 10 evaluations.
